# In mouse chronic pancreatitis CD25$^+$FOXP3$^+$ regulatory T cells control pancreatic fibrosis by suppression of the type 2 immune response

Juliane Glaubitz[1], Anika Wilden[1], Janine Golchert[2], Georg Homuth[2], Uwe Völker [2], Barbara M. Bröker[3], Thomas Thiele[4], Markus M. Lerch[1], Julia Mayerle [1,5], Ali A. Aghdassi[1], Frank U. Weiss[1] & Matthias Sendler [1] ✉

Chronic pancreatitis (CP) is characterized by chronic inflammation and the progressive fibrotic replacement of exocrine and endocrine pancreatic tissue. We identify Treg cells as central regulators of the fibroinflammatory reaction by a selective depletion of FOXP3-positive cells in a transgenic mouse model (DEREG-mice) of experimental CP. In Treg-depleted DEREG-mice, the induction of CP results in a significantly increased stroma deposition, the development of exocrine insufficiency and significant weight loss starting from day 14 after disease onset. In CP, FOXP3$^+$CD25$^+$ Treg cells suppress the type-2 immune response by a repression of GATA3$^+$ T helper cells (Th2), GATA3$^+$ innate lymphoid cells type 2 (ILC2) and CD206$^+$ M2-macrophages. A suspected pathomechanism behind the fibrotic tissue replacement may involve an observed dysbalance of Activin A expression in macrophages and of its counter regulator follistatin. Our study identified Treg cells as key regulators of the type-2 immune response and of organ remodeling during CP. The Treg/Th2 axis could be a therapeutic target to prevent fibrosis and preserve functional pancreatic tissue.

Chronic pancreatitis (CP) is a common gastrointestinal disease, characterized by a progressive inflammatory process of the pancreas which leads to irreversible fibrotic replacement of exocrine and endocrine tissue. These morphologic alterations, which finally cause pancreatic insufficiency[1] and diabetes mellitus are often accompanied by chronic pain[2] and significantly affect the quality of life of patients[3]. As chronic pancreatitis cannot be cured, symptomatic treatment is directed toward relieving pain, improving malabsorption, and treatment of diabetes mellitus[4].

Intrapancreatic activation of digestive enzymes is associated with acinar cell death and represents an initial triggering event in the onset and progression of CP[5–7]. The activation of a pro-inflammatory immune reaction contributes to pancreatic damage[8–10], and cells of the innate immune system, like monocytes/macrophages and neutrophils, infiltrate the damaged pancreas[9–11]. Especially macrophages represent a cell population of the innate immune system with vast plasticity executing different tasks, and therefore are involved in manifestation of acute[9,10] and chronic forms[12,13] of pancreatitis. Recent data illustrate

[1]Department of Medicine A, University Medicine, University of Greifswald, Greifswald, Germany. [2]Interfaculty Institute for Genetics and Functional Genomics, University Medicine Greifswald, Greifswald, Germany. [3]Department of Immunology, Institute of Immunology and Transfusion Medicine, University Medicine, Greifswald, Germany. [4]Department of Transfusion Medicine, Institute of Immunology and Transfusion Medicine, University Medicine, Greifswald, Germany. [5]Department of Medicine II, University Hospital, LMU Munich, Munich, Germany. ✉e-mail: matthias.sendler@uni-greifswald.de

how the disease progresses from pro-inflammation in the acute phase to tissue repair and fibrosis[14,15] in the regeneration phase. Alternatively, activated macrophages are suggested to be the key mediators of fibrogenesis and tissue regeneration[13]. They are characterized by the release of transforming growth factors TGF-β, TGF-α, platelet-derived growth factor PDGF[16] or the anti-inflammatory cytokine IL-10[17] and promote wound healing and fibrogenesis[18]. Quiescent pancreatic stellate cells (PSC) are activated by TGF-β[19] and produce extracellular matrix proteins like type I and type III collagens[20]. Excessive activation of PSC may favor tissue fibrosis over tissue repair and can result in the loss of exocrine and endocrine tissue. In pancreatitis, fibrotic tissue replacement is associated with a type 2 immune response[21,22]. Simultaneously with the pro-inflammatory local response within the damaged pancreas, a systemic counter regulation that prevents hyperinflammation is activated[22].

FOXP3+CD25+ regulatory T cells (Treg cells) are known cellular players balancing the immune response and preventing excessive inflammatory reactions[23] which act on cells of the innate as well as the adaptive immune system[24] and represent regulators of tissue repair processes[25]. CP is characterized by a prominent type 2 immune response[26] involving Th2 cells and alternatively activated macrophages via the IL-4/IL-13 axis[22]. Here, we investigated the effect of FOXP3+CD25+ Treg cells in a mouse model of chronic pancreatitis using DEREG mice that were depleted of regulatory T cells[27].

Here, we show that suppression of the type 2 immune response actioned by Treg cells is indispensable to prevent tissue destruction and pancreatic fibrogenesis.

## Results

### CP is associated with an activation of Treg cells and a type 2 immune response

CP was induced by repetitive caerulein injections over four weeks in C57Bl/6 mice as shown in the treatment scheme (Supplementary Fig. 1a). Animals were sacrificed 3d after the last caerulein treatment. Evaluation of the systemic immune response by flow cytometry analysis of splenocytes showed an increase of GATA3+ Th2 cells (Fig. 1a) in CP mice compared to untreated controls (con), whereas TBET+ Th1 cells and RORγt+ Th-17 cells remained unaffected (Fig. 1b, c). Interestingly we also observed a significant increase of FOXP3+CD25+ regulatory T cells (Fig. 1d). In contrast to these changes of the adaptive immune response, we observed only a slight non-significant increase of CD206+CD11b+Ly6G- macrophages (Fig. 1e). CD163+, a second marker of alternatively activated macrophages in spleen, remained also stable (Fig. 1f). The serum cytokine level of IL-4 was significantly increased in CP mice, whereas the rise of serum levels of IL-10 and IFNγ did not reach significance and TNF as well as IL-17A were not altered (Fig. 1g). To test if CD4+ T-helper cells were the source of these cytokines, we isolated splenocytes and performed flow cytometry analysis of their cytokine production (Fig. 1h). We observed a significant increase in IL-4, IL-13 producing CD4+ cells, and interestingly also IFNγ producing CD4+ cells were significantly elevated in CP mice in contrast to TNF+ CD4+ cells. Beside these Th2 and Th1 cytokine-producing cells, we also observed a slight increase in IL-10+ CD4+ cells. RT-qPCR analysis of splenocytes confirmed an increased production of Th2 cytokines (Fig. 1i).

Next, we were interested if a similar Th2-response, like the one observed in spleen, could also be observed in chronic pancreatitis tissue. Immunofluorescent labeling of CD4 in the mouse pancreas showed infiltration of T cells during CP (Fig. 2a, b). To investigate these CD4+ T cells in more detail we utilized a tissue dissociation kit and analyzed isolated single pancreatic cells by flow cytometry. Again, we observed a significant increase of CD4+ T cells in the pancreas of CP mice (Fig. 2c). We used transcription factor staining to determine more specifically Th1, Th2, Th17, and Treg cells (Fig. 2d). While we observed a high proportion of FOXP3+ Treg cells and GATA3+ Th2 cells, TBET+

Th1 cells, and RORγt+ Th17 cells were much less numerous. Surprisingly the most prominent population of CD4+ T cells were FOXP3+ Treg cells. To confirm these results, we performed staining of Treg cells. CP was also induced in DEpletion of REGulatory T cell (DEREG) mice that express a diphtheria toxin receptor (DTR)-eGFP fusion transgene under the control of the FOXP3 promoter[27]. This allowed us to detect Treg cells by anti-GFP immunofluorescent labeling. Of note, we were able to detect CD3+/GFP positive Treg cells in the pancreas of DEREG mice after induction of chronic pancreatitis (Fig. 2e). Beside Treg cells, we also detected GATA3+ Th2 cells within the pancreas, which are characterized by the release of the Th2 cytokines IL-4 and IL-13. Co-labeling of CD4 and IL-13 showed sporadic double-positive cells within the organ (Fig. 2f). A second marker of Th2 cells is the expression of Prostaglandin D2 receptor 2 (CRTH2 or CD294)[28], which we also found increased in pancreatic tissue of CP mice (Fig. 2g). In addition to CD4+ T cells, a significant amount of CD8α+ cells also migrated into the chronic inflamed pancreas (Fig. 2h).

Most leukocytes in the chronic inflamed pancreas were positive for the chemokine receptor CCR2 (Fig. 3a), suggesting that their migration during CP is triggered by chemokines. CP is also characterized by a prominent increase of alternatively activated CD206+ macrophages which control tissue fibrosis[13]. We observed a high number of CD11b+ alternatively activated macrophages (Fig. 3b). which are positive for CD206 or CD163 (Fig. 3c). Ly6g+ neutrophils could only be detected in negligible numbers (Supplementary Fig. 2). Histologic labeling and cell quantification of mouse CP tissue also showed increased numbers of alternatively activated CD206+ macrophages accompanied by prominent collagen 1 staining, which indicates fibrotic scarring (Fig. 3d). These CD206+ macrophages express the cytokine IL-33, as shown by fluorescent labeling (Fig. 3e). IL-33 is involved in the differentiation of Th2 cells and of innate lymphoid cells type 2 (ILC2) via the ST2-receptor. ILC2 are a major component of the type 2 immune response. Innate lymphoid cells are characterized by the absence of specific linage markers (lin-) but express CD45, CD127 and CD90 (Supplementary Fig. 3). ILC2s additionally express the transcription factor GATA3. We could identify a significant increase of ILC2s (lin-CD45+CD127+CD90+GATA3+) in the pancreas of CP mice (Fig. 3f). Like Th2 cells, ILC2s can also express cytokines IL-4[29] and IL-13[30]. Mean fluorescent intensity of IL-4 and IL-13 staining was elevated in flow cytometry analysis of splenic GATA3+ ILCs (ILC2s) compared to GATA3- ILCs or isotype controls (Supplementary Fig. 4a). In addition, GATA3+CD4- cells in CP tissue were detected positive for the cytokines IL-4 and IL-13 by immunofluorescence labeling (Supplementary Fig. 4b), which suggests that ILC2s are involved in pancreatic fibrogenesis.

The type 2 immune responses are known to regulate tissue repair and fibrosis. Azan blue staining illustrates the increased fibrosis in pancreatic tissue of CP mice (Fig. 3g). Pancreatic fibrosis is mainly mediated by increased numbers of pancreatic stellate cells (PSCs), which can be detected in CP tissue by the expression of CD271 or GFAP (Fig. 3h). We also observed increased levels of soluble collagen in serum of CP mice (Fig. 3i) and demonstrated CP-dependent changes of pancreatic tissue composition by Masson Goldner trichrome staining (Fig. 3j, k). In CP, exocrine tissue (red) is replaced by fibrotic tissue (green). The yellow-marked free space indicates edema of fatty tissue which arises during CP. Our histomorphological analysis showed a significant loss of acinar cells accompanied by an increase of fibrotic tissue (Fig. 3l).

### Blockage of Th2 cell response mitigates tissue fibrosis

It is known that the type 2 immune response is controlled by the IL-4/IL-13 - STAT6 signaling pathway. Th2 differentiation and M2 polarization are regulated by IL-4/IL-13, and therefore contribute to fibrosis. To investigate the effect of IL-4 during chronic pancreatitis, we treated C57Bl/6 mice with an IL-4 neutralizing antibody, whereas control

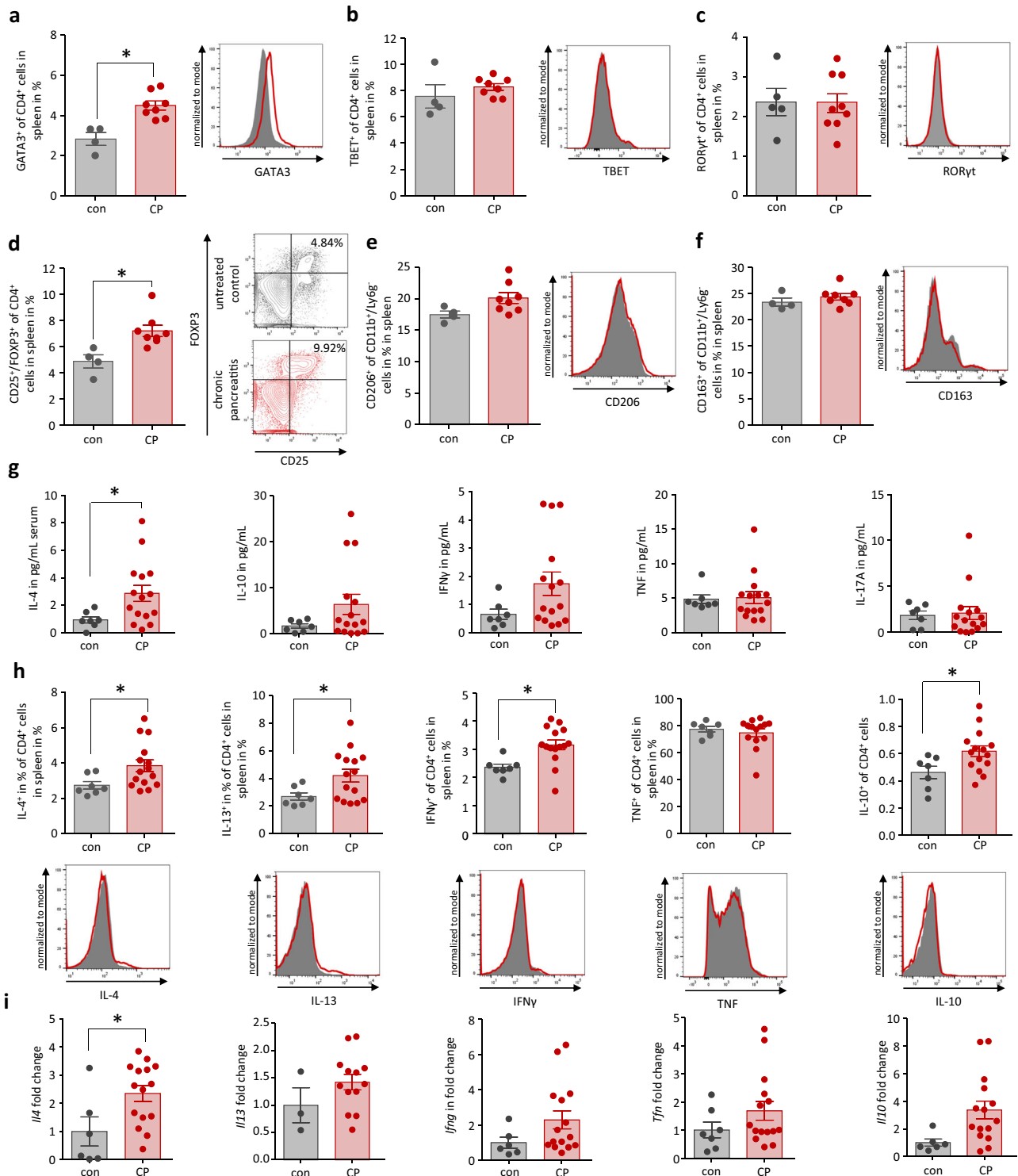

**Fig. 1 | Systemic immune response during chronic pancreatitis in mice.** Splenocytes were analyzed from control (con) and chronic pancreatitis (CP) mice. **a** T cell (CD4⁺) differentiation is shown by intracellular transcription factor staining of GATA3 for Th2 cells ($p = 0.0016$, con $n = 4$/CP $n = 8$), **b** TBET for Th1 cells, **c** RORγt for Th17 cells and **d** FOXP3 in combination with CD25 to discriminate Treg cells ($p = 0.0085$, con $n = 4$/CP $n = 8$). **e**, **f** In contrast to GATA3⁺ Th2 cells an increase of CD206⁺ or CD163⁺ alternatively activated macrophages could not be observed. **g** Serum concentration of IL-4 was significantly increased ($p = 0.0422$, con $n = 7$/CP $n = 15$), whereas the rise of IL-10, IFNγ, TNF, and IL-17A concentrations were not significant (con $n = 7$/CP $n = 15$). **h** The intracellular cytokine production of CD4⁺ T cells was analyzed by labeling for IL-4, IL-13, IFNγ, TNF, and IL-10 after brefeldin a/monensin treatment. Th2 cytokines IL-4 ($p = 0.0443$, con $n = 7$/CP $n = 15$), IL-13

($p = 0.0455$, con $n = 7$/CP $n = 15$) and IL-10 ($p = 0.0294$, con $n = 7$/CP $n = 15$) were elevated in CD4⁺ T cells, whereas of Th1 cytokines only IFNγ ($p = 0.0072$, con $n = 7$/CP $n = 15$) was elevated but not TNF (con $n = 7$/CP $n = 14$). **i** Gene expression in splenocytes was also analyzed by RT-qPCR for *Il4* ($p = 0.0250$, con $n = 7$/CP $n = 15$), *Il13* (con $n = 3$/CP $n = 13$), *Il10* (con $n = 6$/CP $n = 15$), *Tnf* (con $n = 7$/CP $n = 15$) and *Ifng* (con $n = 6$/CP $n = 15$) and confirmed the Th2 differentiation. Transcript levels as determined by RT-qPCR were normalized using *Rn5s* as internal calibrator gene and were related to the corresponding mRNA amounts in control mice. All data were presented as means ± SEM, statistically significant differences were tested by unpaired two-tailed students t-test for independent samples and significance levels of $p < 0.05$ are marked by an asterisk (**a**, **d**, **g**, **h**, **i**). Source data are provided as a Source Data file.

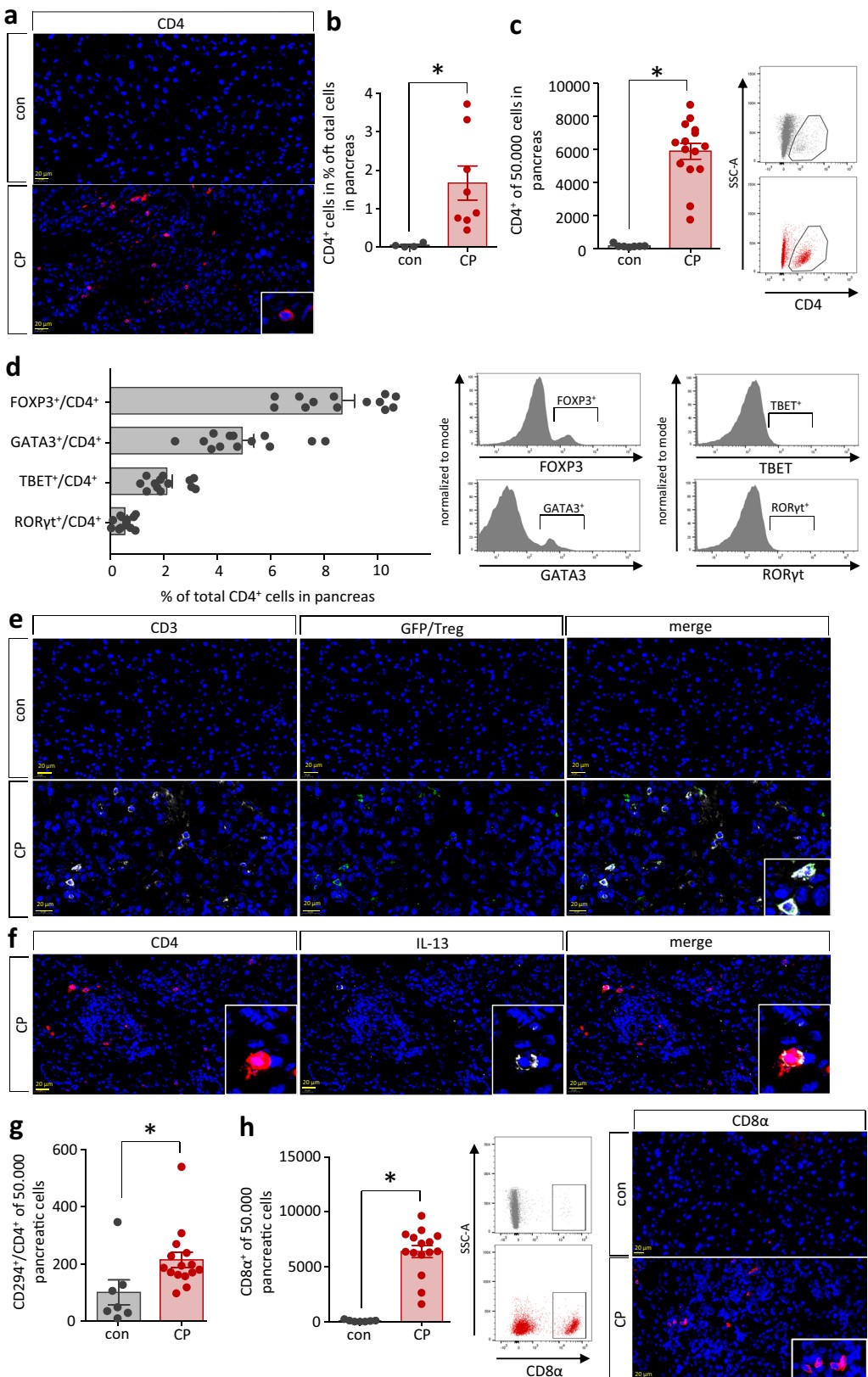

animals received isotype antibody in the same treatment scheme (Supplementary Fig. 1b). Chronic pancreatitis was induced as previously described. Treatment with anti-IL-4 did not result in weight changes over the period of CP (Supplementary Fig. 5a). Flow cytometry analysis of splenocytes from anti-IL-4 treated mice showed a significant reduction of GATA3⁺ Th2 cells and of GATA3⁺ ILC2s (lin⁻

CD45⁺CD127⁺CD90⁺), whereas TBET⁺ Th1 cells, FOXP3⁺CD25⁺ Treg cells as well as TBET⁺ ILC1s were not affected by the anti-IL-4 treatment (Supplementary Fig. 5b, c). Numbers of CD4⁺ T cells and especially of CD4⁺/STAT6⁺ cells were significantly reduced in the pancreas of these CP mice (Supplementary Fig. 5d, e). The pancreas histology showed differences between anti-IL-4 and isotype-treated mice, including a

**Fig. 2 | T cells infiltrate chronic inflamed pancreatic tissue.**
**a**, **b** Immunofluorescent labeling of CD4 in pancreas of CP mice showed a significant infiltration of CD4$^+$ T cells ($p = 0.0308$, con $n = 4$/CP $n = 8$), scale bars represent 20 μm. **c** Flow cytometry analysis of isolated pancreatic leukocytes confirmed the observed significant increase of CD4$^+$ T cells ($p < 0.0001$, con $n = 7$/CP $n = 15$). **d** The largest proportion of T cells were FOXP3$^+$ Treg cells followed by GATA3$^+$ Th2 cells, whereas TBET$^+$ Th1 and RORγt$^+$ Th17 cells could only be detected in small amounts ($n = 13$). **e** Analysis of DTR-eGFP transgene expression in DT-treated DEREG mice and untreated control animals (con) demonstrated presence of CD3$^+$/GFP$^+$ cells exclusively in pancreatic sections of CP animals, scale bars represent 20 μm. **f** Labeling of CD4 in combination with the Th2 cytokine IL-13, identified double positive Th2 cells in pancreas of CP mice, scale bars represent 20 μm. **g** Elevation of CRTH2 (CD294), a third marker of Th2 cells, was measured by flow cytometry analysis of pancreatic leukocytes and underlines the increase of Th2 cells within the pancreas ($p = 0.0326$, con $n = 7$/CP $n = 15$). **h** An infiltration of CD8α cells into the pancreas could also be observed by flow cytometry ($p < 0.0001$, con $n = 7$/CP $n = 15$) and immunofluorescent labeling, scale bars represent 20 μm. All data were presented as means ± SEM, statistically significant differences were tested by unpaired two-tailed students $t$-test for independent samples and significance levels of $p < 0.05$ are marked by an asterisk (**b**, **c**, **g**, **h**). Source data are provided as a Source Data file.

markedly reduced fibrosis in the anti-IL-4 treated CP mice, as is shown by Masson Goldner trichrome and azan blue staining. Immunofluorescent labeling of αSMA, a marker for activated stellate cells, is also reduced in these animals, whereas the number of α-amylase positive acinar cells was increased (Supplementary Fig. 5f). Quantification of fibrosis and αSMA$^+$ cells showed a significant reduction in the group of anti-IL-4 treated animals (Supplementary Fig. 5g). Immunofluorescent labeling of collagen 1 and CD206 confirmed reduced collagen production in the pancreas of anti-IL-4 treated mice, whereas the number of CD206$^+$ cells was not affected (Supplementary Fig. 5g).

It is known that IL-4/IL-13 are responsible for the formation of fibrosis during CP[13]. Beside Th2 cells (Figs. 1h and 2f) and ILC2s (Supplementary Fig. 4), pancreatic stellate cells are another source of IL-4[13]. To consider the influence of Th2/ILC2s on fibrogenesis in more detail we performed a blockade of CRTH2 by its antagonist OC000459. Blockage of the Th2/ILC2s response was achieved by a daily application of 30 μg of the CRTH2 antagonist OC000459[31], whereas control animals received vehicle only (0.625% DMSO) as previously described by Wojno et al.[32] (Supplementary Fig. 1c). OC000459 is a highly potent and selective CRTH2 antagonist that inhibits PGD2-mediated activation of eosinophils, Th2 cells, and ILC2s. The different treatment conditions did not cause significant weight loss over the time of CP development (Fig. 4a). Quantification of CD4$^+$ cells showed a significant reduction of CD4$^+$ T cell numbers in the pancreas, and also CD4$^+$STAT6$^+$ Th2 cells were markedly reduced in the OC000459 treated group (Fig. 4b, c). Immunofluorescent labeling also showed a significant reduction of CD206$^+$ macrophages in the pancreas of OC000459 treated mice (Fig. 4d, e). Pancreatic amylase expression was increased and the PSC marker αSMA showed a decreased number of PSCs in the pancreas of OC000459 treated animals (Fig. 4f, g). Histomorphological analysis of Masson Goldner trichrome staining confirmed a significantly decreased tissue fibrosis in the OC000459 treated group (Fig. 4h). H&E and azan blue staining as well as immunohistochemical labeling of Collagen 1 supported this observation (Fig. 4i). Furthermore, gene expression analysis of pancreatic tissue showed a significantly decreased expression of the *Mrc1* gene encoding for CD206, the *Col1a1* gene and *Il4* encoding for the cytokine IL-4 (Fig. 4j). The decrease of the cytokine IL-13 expression did not reach significance, but other genes of type 2 related cytokines and growth factors like *Il10*, *Areg* or *Tgfb* were significantly reduced in the OC000459 treated group (Fig. 4j).

### Treg cells suppress the splenic type 2 immune response during CP

Beside Th2 cells, Treg cells were the most prominent T cell population, which we identified in spleen as well as in pancreas of CP mice, so we next investigated the impact of regulatory T cells during CP. In DEREG mice, Treg cells are selectively depleted by diphtheria toxin (DT) administration because these mice express the gene encoding a diphtheria toxin receptor DTR-eGFP under the *Foxp3* promoter[27]. A DEREG control group received phosphate-buffered saline (PBS) instead of DT (Supplementary Fig. 1d). The success of Treg depletion was analyzed in lymph node sections by labeling the GFP$^+$ Treg cells with an anti-GFP antibody and in spleen cell suspensions by flow cytometric analysis of GFP$^+$CD4$^+$ cells (Figs. 5a, b). In both organs, Treg cells were strongly reduced after DT injection. Increased levels of CD69, a general activation marker of CD4$^+$ T cells, indicated significant T-effector cell activation in DT-treated mice (Fig. 5c). The polarity of the T-effector cell response (Th1/Th2/Th17 balance) was studied by flow cytometric analysis of the expression of the transcription factors GATA3, TBET and RORγt. Treg depletion during CP induction resulted in almost tripled the percentage of GATA3$^+$ Th2 cells in the spleen (Fig. 5d) but did not change the Th1- or Th17-response (Fig. 5e, f). Corroborating this finding, IL-4 and IL-10, cytokines that accompany a Th2/type 2 immune response, were significantly elevated in the serum of DT-treated mice. On the other hand, pro-inflammatory cytokines like IL-6 were not significantly affected by the depletion of FOXP3$^+$ Treg cells (Fig. 5g). In line with the elevated Th2 response we observed an increased number of CD11b$^+$CD206$^+$ macrophages in the spleen of DT-treated DEREG mice (Fig. 5h). Of interest was also the observation that CD8α$^+$ T cells were significantly increased in DT-treated DEREG mice during CP (Fig. 5i).

### Treg cells suppress the pro-fibrotic pancreatic type 2 immune response in CP

The systemic type 2 immune response is suppressed by FOXP3$^+$CD25$^+$ Treg cells and therefore significantly increased in their absence. Labeling CD4$^+$ T cells in pancreatic tissue sections demonstrated that Treg depletion intensifies the pancreatic T cell infiltration (Fig. 6a, b). Similarly, CD8α$^+$ cells were significantly increased in numbers in the DT-treated CP mice (Fig. 6c). In line with these observations, also the number of CD206$^+$ macrophages were found significantly increased in sections of DT-treated animals with CP (Fig. 6d). Flow cytometry of isolated pancreatic leukocytes (Fig. 6e) and gene expression analysis of the alternative activated macrophage markers MRC-1 and YM1 in pancreatic tissue (Fig. 6f) and confirmed the increase of CD206$^+$ macrophages in DT-treated mice. Expression of *Il10* encoding cytokine IL-10 that is mainly released by CD206$^+$ macrophages was increased in pancreatic tissue of DT-treated mice with CP. In contrast, the gene expression of the M1 macrophage-related cytokine IL-1β was not affected by the depletion (Fig. 6f). Macrophage differentiation towards alternatively activated anti-inflammatory M2-like phenotype correlates with an increased number as well as activation of pancreatic stellate cells. Labeling of α smooth muscle actin (αSMA) demonstrated increased numbers of activated stellate cells in the pancreas of Treg-depleted animals accompanied by increased expression of *Col1a1* encoding the extracellular matrix protein collagen 1 (Fig. 6g). Flow cytometric analysis of cells isolated from CP tissue of DEREG mice demonstrated that systemic depletion of Treg cells during CP resulted in a significant increase of cells positive for the stellate cell markers CD271 and GFAP (Fig. 6h, i). Additional gene expression analysis of αSMA confirmed this finding (Fig. 6j). Treg depletion enhanced pancreatic fibrosis which is visible in histological staining with Masson Goldner or Azan blue (Fig. 6k). The fibrotic cell area in relation to the total tissue area is significantly enlarged in Treg-depleted mice (Fig. 6i, m). A significant reduction of exocrine tissue area (surrounded in red)

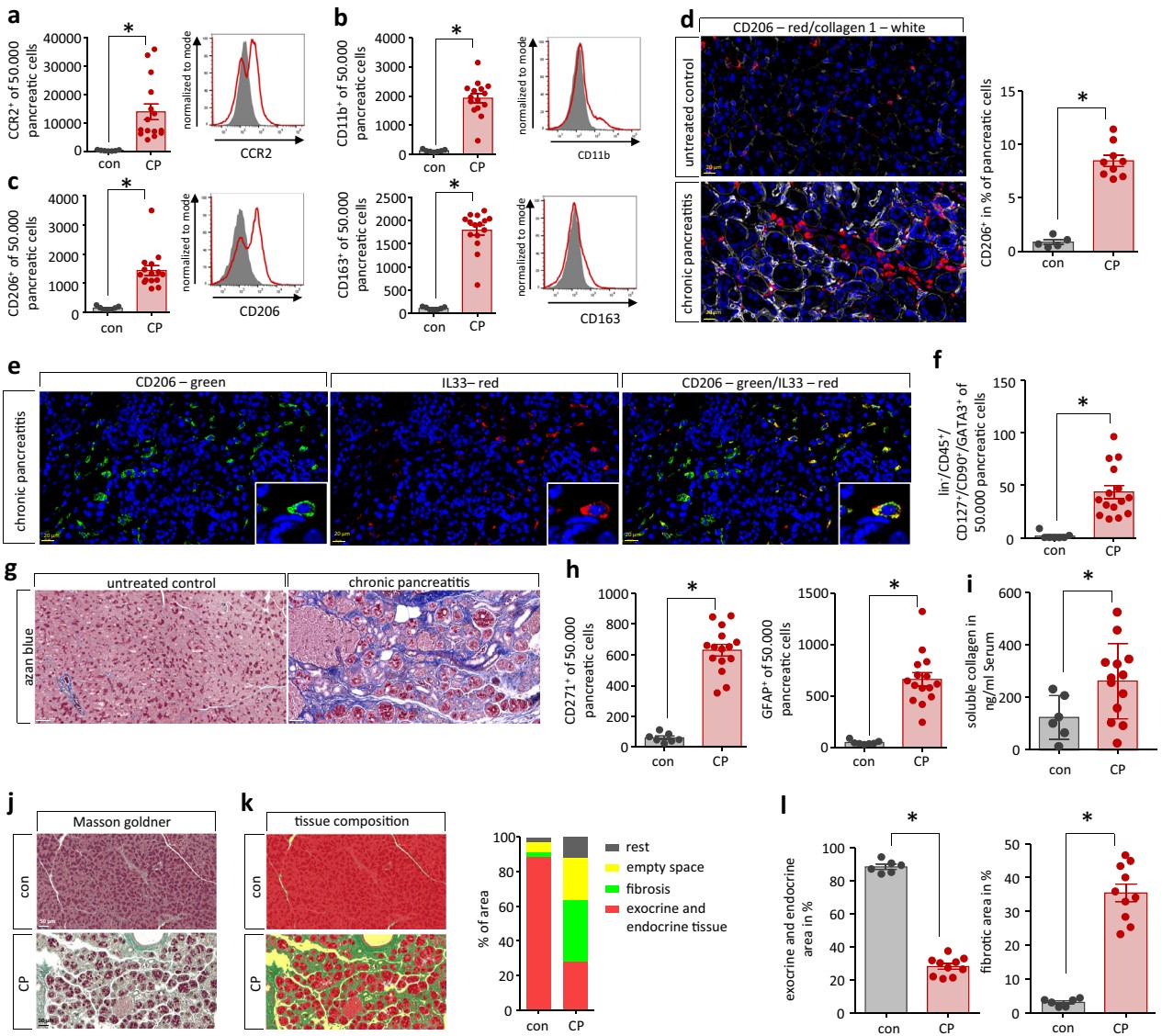

**Fig. 3 | Chronic pancreatitis leads to pancreatic tissue remodeling. a** Flow cytometry analysis of isolated leukocytes from pancreatic tissue showed a significant increase of CCR2⁺ cells ($p = 0.0032$, con $n = 7$/CP $n = 15$). **b** CD11b⁺ labeling identified that also cells of the innate immune system infiltrate into the damaged pancreas ($p < 0.0001$, con $n = 7$/CP $n = 15$). **c** Most of these cells were positive for CD206 ($p < 0.0001$, con $n = 7$/CP $n = 15$) or CD163 ($p < 0.0001$, con $n = 7$/CP $n = 15$), which characterized them as alternatively activated macrophages. **d** Labeling of CP tissue for CD206 illustrates their increase in pancreatic tissue of mice. A quantative analysis showed that ~9% of all cells were positive for CD206 in CP tissue ($p < 0.0001$, con $n = 5$/CP $n = 9$), scale bars represent 20 μm. **e** IL-33 expression of CD206⁺ macrophages was demonstrated by immunofluorescent labeling, scale bars represent 20 μm. **f** This cytokine is a strong inducer of ILC2s (lin⁻CD45⁺CD127⁺CD90⁺GATA3⁺) which were also significantly elevated in the pancreas of CP mice ($p = 0.0002$, con $n = 7$/CP $n = 15$). **g** The replacement of exocrine pancreas by fibrotic tissue in CP was illustrated by azan blue staining of the pancreas,

scale bars represent 50 μm. **h** Flow cytometry analysis of isolated pancreatic cells for PSC marker proteins CD271 ($p < 0.0001$, con $n = 7$/CP $n = 15$) and GFAP ($p < 0.0001$, con $n = 7$/CP $n = 15$) identified a significant increase. **i** Serum collagen levels were significantly elevated in CP ($p = 0.0433$, con $n = 7$/CP $n = 13$) and collagen deposition and tissue fibrosis were observed in histology. **j** Masson Goldner trichrome staining was utilized to evaluate the tissue remodeling during CP, scale bars represent 50 μm. **k** Area quantification (in%) by pattern quant software illustrates the loss of exocrine and endocrine tissue (red) and the increase of fibrotic area (green) and of free space, which reflects fatty tissue and edema (yellow). **l** Area quantification of exocrine and endocrine active tissue ($p < 0.0001$, con $n = 6$/CP $n = 10$) versus fibrosis ($p < 0.0001$, con $n = 6$/CP $n = 10$) demonstrated the significant changes in organ architecture. All data were presented as means ± SEM, statistically significant differences were tested by unpaired two-tailed students t-test for independent samples and significance levels of $p < 0.05$ are marked by an asterisk (**a**–**f**, **h**, **i**, **l**). Source data are provided as a Source Data file.

was detected in pancreatic sections from DT-treated animals (Supplementary Fig. 6a). Immunofluorescent labeling with anti-cytokeratin 19 antibody indicated increased numbers of ductal structures, suggesting an increased acinar to ductal metaplasia in the DT-treated mice (Supplementary Fig. 6b).

To analyse if the permanent depletion of Treg cells has a direct effect on the animals, DEREG mice were treated with DT or PBS over the same period of time, but without inducing CP. After 6 days we

detected almost no GFP⁺CD25⁺ Treg cells in the spleens of DT-treated animals, but after 28d a small DT-resistant population of Treg cells could be observed (Supplementary Fig. 7a and 7b), as described in the literature. Genetic defects in the *Foxp3* or *CTLA4* gene cause severe autoimmune diseases[33], but a depletion in adult animals does not provoke similar disease phenotypes[34]. The long-term depletion of Treg cells did not affect the effector T cell response in these animals (Supplementary Fig. 7c) and we observed no chronic inflammation or organ

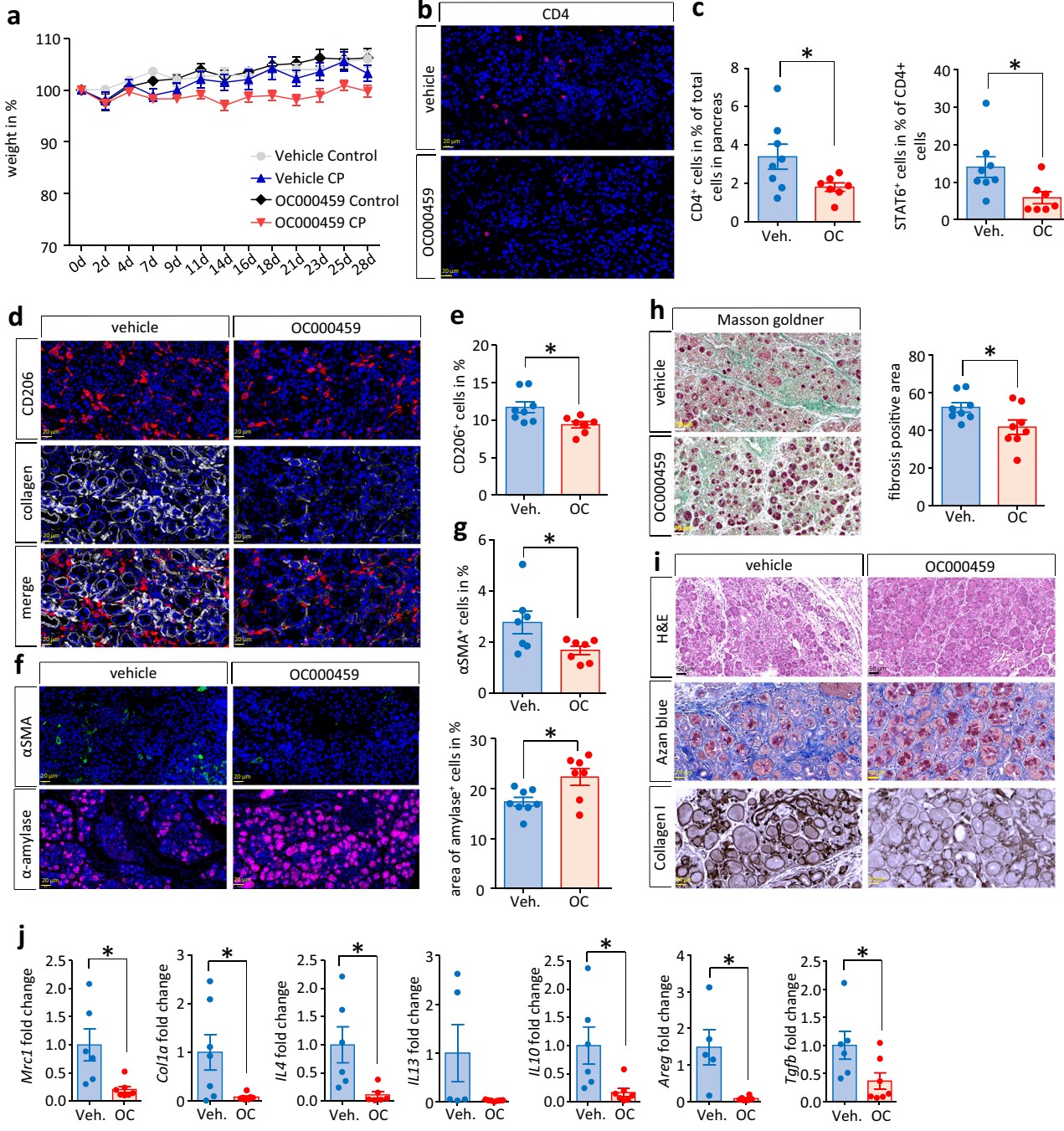

**Fig. 4 | Treatment with OC000459, a selective CRTH2 antagonist, mitigates the type 2 immune response.** CP was induced in C57Bl/6 mice by repetitive caerulein injections over 4 weeks. In addition, animals received daily 30 μg of OC000459 or vehicle (0.625% DMSO). **a** Body weight changes over time were not significant between all groups (vehicle control n = 3/vehicle CP n = 8/OC000459 control n = 4/OC000459 CP n = 8). **b** OC000459 significantly reduced the CD4⁺ T cell infiltration of the pancreas in CP mice (p = 0.0469, Veh. n = 8/OC n = 7), scale bars represent 20 μm. **c** The proportion of CD4⁺STAT6⁺ cells was also smaller in CP tissue of OC000459 treated mice (p = 0.0285, Veh. n = 8/OC n = 7). **d, e** Labeling of CD206⁺ showed a significant reduction of alternatively activated macrophages in the pancreas (p = 0.0174, Veh. n = 8,/OC n = 7), scale bars represent 20 μm. **d, f** Immunofluorescence images showed a decreased labeling of collagen 1 and αSMA, whereas the amount of α-amylase in the pancreas of OC000459 treated mice was increased, scale bars represent 50 μm. **g** Quantification of immuno-fluorescence signals for αSMA⁺ cells showed a significant decrease (p = 0.0414, Veh. n = 7/OC n = 7), whereas the area of amylase⁺ cells was significantly increased in the pancreas of OC000459 treated mice (p = 0.0174, Veh. n = 8/OC n = 7).

**h** Quantification by pattern quant software showed significantly less fibrotic tissue in OC000459 treated mice (p = 0.0397, Veh. n = 8/OC n = 8). **i** H&E staining, azan blue staining and immune labeling of collagen 1 underlines the reduced fibrosis in the OC000459 treated group, scale bars represent 50 μm. **j** Gene expression analysis of pancreatic tissue by RT-qPCR demonstrated significantly decreased transcript levels for *Mrc1* (p = 0.0130, Veh. n = 6/OC n = 7), *Col1a* (p = 0.0273, Veh. n = 7/OC n = 7), *Il4* (p = 0.0212, Veh. n = 6/OC n = 6), *Il10* (p = 0.0226, Veh. n = 6/OC n = 7), *Areg* (p = 0.0104, Veh. n = 5/OC n = 6) and *Tgfb* (p = 0.0433, Veh. n = 6/OC n = 7), indicating a reduced type 2 immune response. Transcript levels of *Il13* (Veh. n = 5/OC n = 7) were reduced but the decrease did not reach significance. Transcript levels as determined by RT-qPCR were normalized using *Rn5s* as internal calibrator gene and were related to the corresponding mRNA amounts in control mice. All data were presented as means ± SEM, statistically significant differences were tested by unpaired two-tailed students t-test for independent samples and significance levels of p < 0.05 are marked by an asterisk (**c**, **e**, **h**, **g**, **j**). Source data are provided as a Source Data file.

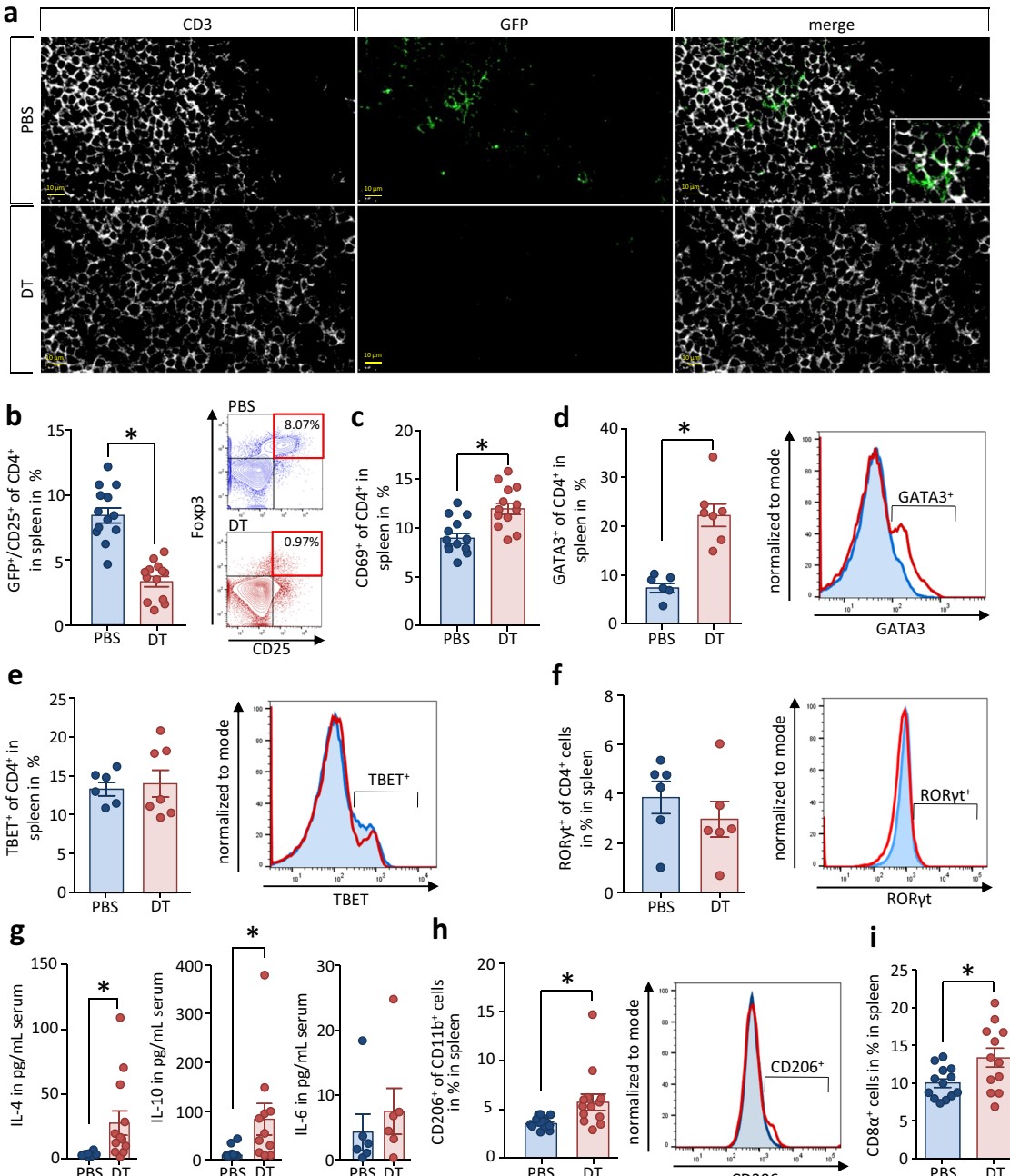

**Fig. 5 | Treg cells suppress a systemic type 2 immune response.** CP was induced in DEREG mice by repetitive caerulein i.p. injection over 4 weeks in Treg-depleted animals. Control mice received PBS instead of DT. **a** An anti-GFP antibody was used for immunofluorescence staining of GFP⁺ Treg cells in lymph node, scale bars represent 10 μm. **b** Determination of GFP⁺ and CD25⁺ splenocytes was performed by flow cytometry ($p < 0.0001$, PBS $n = 13$/DT $n = 13$). **c** T-effector cell activation was measured by detection of CD69 on CD4⁺ T cells ($p = 0.0008$, PBS $n = 13$/DT $n = 13$). **d–f** Furthermore, the T effector differentiation of CD4⁺ splenocytes were quantified by flow cytometry. The T effector immune response was measured by intracellular staining of GATA3 ($p = 0.0002$, PBS $n = 6$/DT $n = 7$), TBET (PBS $n = 6$/DT $n = 7$) and RORγt (PBS $n = 6$/DT $n = 6$) in CD4⁺ cells. Detected levels of the transcription factors TBET and RORγt were not significantly different between both groups. In contrast to the Th1 and Th17 immune responses, the Th2 immune response, as hallmarked by the presence of GATA3, was increased by depletion of regulatory T cells during CP. **g** Measurement of serum cytokines by a cytometric bead array showed an increase of type 2 cytokines like IL-10 ($p = 0.0271$, PBS $n = 12$/DT $n = 11$) and IL-4 ($p = 0.0127$, PBS $n = 12$/DT $n = 12$), whereas typical type 1 cytokines like IL-6 (PBS $n = 6$/DT $n = 6$) were not affected. **h** Analysis of the surface marker CD206 on CD11b⁺ cells in the spleen by flow cytometry indicates that not only the Th2 response was increased but also the number of alternatively activated anti-inflammatory macrophages ($p = 0.0186$, PBS $n = 13$/DT $n = 13$). **i** The number of CD8α⁺ T cells is also significantly elevated in the absence of Treg cells during CP ($p = 0.0206$, PBS $n = 13$/DT $n = 12$). All data were presented as means ± SEM, statistically significant differences were tested by unpaired two-tailed students t-test for independent samples and significance levels of $p < 0.05$ are marked by an asterisk (**c**, **d**, **g**, **h**, **i**). Source data are provided as a Source Data file.

damage based on H&E staining analysis of kidney, lung, liver, and small intestine; all organs were unaffected by the Treg depletion (Supplementary Fig. 7d). It has been reported that the genetic deletion or neonatal depletion of Treg cells resulted in a complete destruction of pancreatic tissue by infiltrating T cells[27,33,34]. We observed in two out of five animals small, isolated foci of infiltration by CD3⁺ or CD8⁺ T cells in the pancreas of DT-treated animals, but no general tissue destruction or CD206⁺ activation and increased fibrosis (Supplementary Fig. 8a

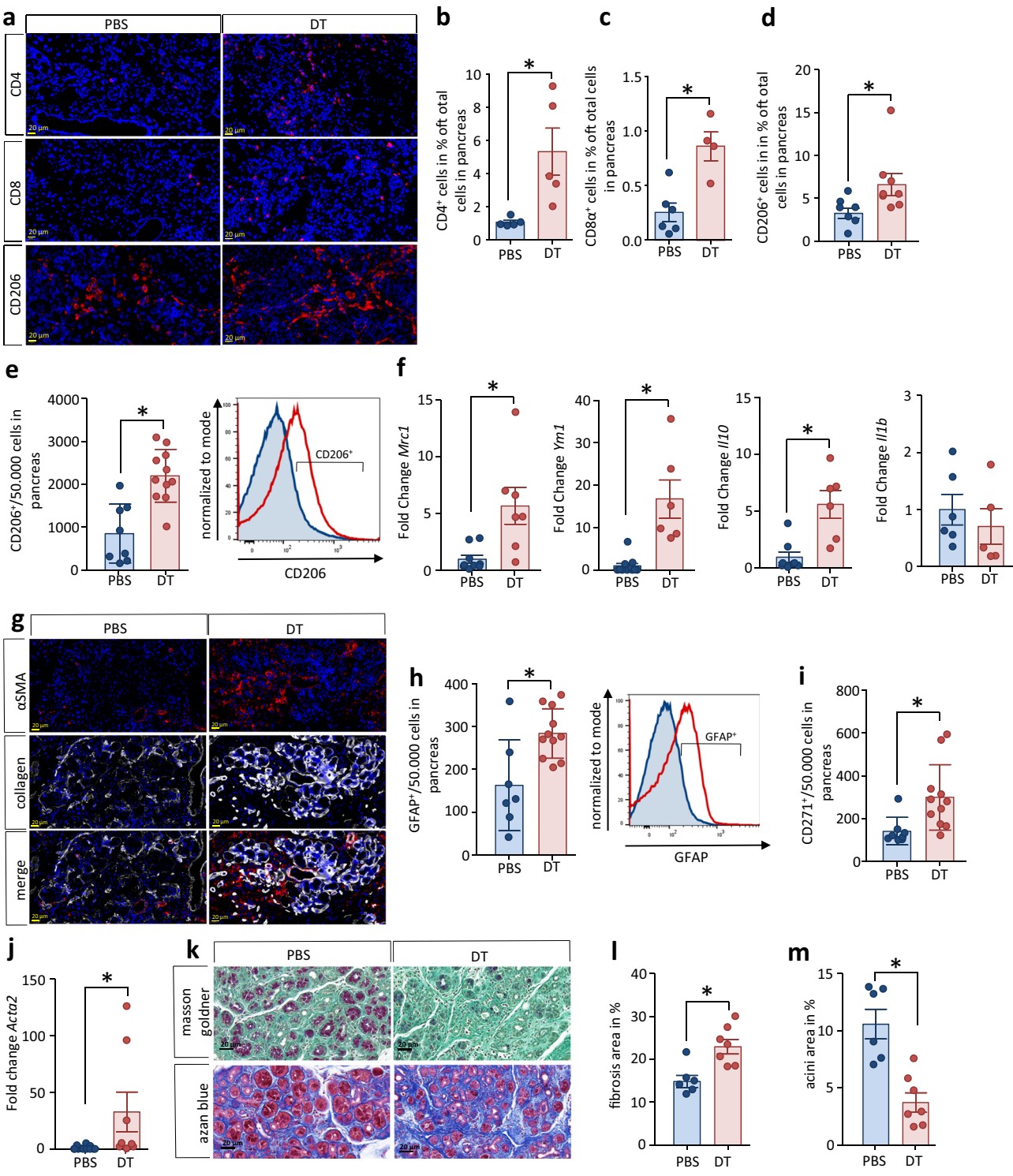

and 8b). The major part of the pancreas was unaffected and unchanged serum lipase levels of DT-treated animals excluded that infiltrating T cells might have triggered the development of acute pancreatitis (Supplementary Fig. 8c).

### Treg cells regulate organ remodeling during CP

Aside from fibrotic tissue replacement, acinar cell regeneration is also involved in the process of organ remodeling during CP. As macrophages are known to influence organ regeneration[15], we investigated the release of growth factors and growth factor-related proteins from them in an in vitro co-culture model of bone marrow-derived

macrophages (BMDM) and CCK stimulated acini, as previously described[10]. After 6 h of co-culture, the BMDMs were carefully washed, total RNA was prepared and subsequently used for transcriptome analyses using Affymetrix arrays. We observed increased mRNA levels of genes encoding various growth factors like nerve growth factor (NGF), inhibin βa (INHBA), fibroblast growth factor 1 (FGF1), Amphiregulin (AREG) but also of counter regulators like follistatin (FST) (Fig. 7a).

Next, we studied organ regeneration by analyzing the gene expression of these growth factors and their inhibitors by RT-qPCR using RNA prepared from the pancreas of DEREG mice with CP.

**Fig. 6 | Depletion of Treg cells strengthen the pancreatic remodeling during CP.**
**a**–**d** Quantification of infiltrating immune cells by evaluation of immuno-fluorescence labeling of CD4 ($p = 0.0172$, PBS $n = 5$/DT $n = 5$), CD8α ($p = 0.0035$, PBS $n = 6$/DT $n = 4$) and CD206 ($p = 0.0451$, PBS $n = 7$/DT $n = 8$) shows that their numbers were significantly elevated in DT-treated animals, scale bars represent 20 μm. **e** Flow cytometry analysis of leukocytes isolated from pancreatic tissue provides evidence for increased alternatively activated CD206+ M2 macrophages in DT-treated mice ($p = 0.0003$, PBS $n = 8$/DT $n = 11$). **f** Gene expression analysis by RT-qPCR confirmed elevated expression of *Mrc1* ($p = 0.0064$, PBS $n = 9$/DT $n = 7$) and *Ym1* ($p = 0.0003$, PBS $n = 11$/DT $n = 6$) encoding anti-inflammatory macrophage markers, increased transcript levels for the anti-inflammatory cytokine IL-10 ($p = 0.0020$, PBS $n = 8$/DT $n = 6$) whereas mRNA amounts of the pro-inflammatory IL-1β cytokine (PBS $n = 6$/DT $n = 5$) underlines this observation. Transcript levels as determined by RT-qPCR were normalized using *Rn5s* as internal calibrator gene and were related to the corresponding mRNA amounts in control mice. **g** M2-like macrophages in DT-treated mice trigger pancreatic stellate cells (PSC), as shown by staining of αSMA and collagen 1, scale bars represent 20 μm. **h**, **i** Flow cytometry analysis of dissociated pancreatic tissue shows a significant higher number of GFAP+ ($p = 0.0059$, PBS $n = 7$/DT $n = 11$) and CD271+ cells ($p = 0.0140$, PBS $n = 8$/DT $n = 11$), **j** and gene expression analysis of αSMA confirms activation of PSC after depletion of Treg cells during CP ($p = 0.0373$, PBS $n = 12$/DT $n = 8$). **k** Finally, the higher PSC number causes increased fibrosis and pronounced loss of acinar cells, scale bars represent 100 μm. **l**, **m** Quantitative analyses performed by Masson Goldner demonstrate a significant increase in fibrosis ($p = 0.0042$, PBS $n = 6$/DT $n = 7$) accompanied by decreased acinar cell numbers ($p = 0.0008$, PBS $n = 6$/DT $n = 7$) in the absence of Treg cells. All data were presented as means ± SEM, statistically significant differences were tested by unpaired two-tailed students *t*-test for independent samples and significance levels of p < 0.05 are marked by an asterisk (**b**–**f**, **h**–**j**, **l**, **m**). Source data are provided as a Source Data file.

In the pancreas of DT-treated Treg depleted mice, there was elevated expression of the INHBA encoding the inhibin encoding subunit β$_A$ subunit of the growth factor activin. The transcription of INHA that encoding the inhibin α subunit was not affected (Fig. 7b). Activin is a homodimer of the inhibin β$_A$ subunit, whereas its inhibitory counterpart inhibin represents a heterodimer of inhibin α and inhibin β$_A$. Our results suggest an increased formation of homo-dimeric activin at the expense of the heterodimeric inhibin. Concomitantly the activin counter regulator follistatin was downregulated in the pancreas of DT-treated mice (Fig. 7a). These results are in line with the results of our in vitro co-culture experiments (Fig. 7b). Immunofluorescent labeling of CD206+ macrophages and of the inhibin βA subunits demonstrated co-localization of both markers in fibrotic areas of the pancreas with increased fluorescence signals in the absence of Treg cells (Fig. 7c). The expression of FGF1 and TGF-β were also elevated in the pancreas of these mice (Fig. 7d). Both are known to activate pancreatic stellate cells[19], and to stimulate the production of extracellular matrix proteins like collagens. Collagen 1 which was also significantly elevated both at the mRNA (Fig. 7d) and protein levels (Fig. 7c). The immune labeling analysis of fibroblast growth factor receptor showed co-localization with αSMA expressing PSCs (Fig. 7e). Our results show that in CP regulatory T cells suppress the excessive release of growth factors by pancreatic macrophages, thereby limiting the fibrotic replacement of exocrine pancreatic tissue.

### Treg cells suppress innate lymphoid cells type 2
Besides Th2 cells and alternatively activated macrophages ILC2 are involved in the type 2 immune response. We, therefore, reasoned that ILC2s might also be regulated by Treg cells during CP. First we investigated ILC2s in the spleen of DEREG mice with CP. The absence of Treg cells did not result in an increase of total ILCs in the spleen (Fig. 8a). When we discriminated ILCs via the expression of the transcription factors TBET and GATA3 into ILC subpopulations of type 1 and type 2 (Fig. 8b), comparable to the differentiation of T cells, we observed a shift towards ILC2 differentiation, similar to the increased Th2 differentiation. Whereas the numbers of TBET+ ILC1s did not change after DT treatment, those of GATA3+ ILC2s increased significantly (Fig. 8c, d). ILC2s are known to drive tissue fibrosis[30,35]. Therefore, we next asked whether ILCs were present locally in pancreatic tissue during CP and whether these were also regulated by Treg cells. We were able to detect lin−CD45+CD127+CD90+ cells by flow cytometry analysis of isolated pancreatic leukocytes (Fig. 8e). Animals suffering from CP showed an increase of ILCs in the pancreas that was significantly enhanced in DT-treated mice. The ILCs that are significantly increased in the absence of Treg cells strongly express GATA3, which identifies them as ILC2s (Fig. 8f). ILC2 are characterized by the expression of amphiregulin (AREG), a transmembrane glycoprotein that belongs to the epidermal growth factor family. Expression analysis of the encoding AREG gene in pancreas tissue by quantitative RT-qPCR indicated a significant increase in DT-treated mice. A third line of evidence for the involvement of ILCs comes from immunofluorescence analysis experiments detecting both CD90 and AREG, which are found co-expressed by ILC2s. Pancreatic tissue sections illustrate the localization of high numbers of ILC2s in the chronically inflamed pancreas of DT-treated DEREG mice (Fig. 8g). We also evaluated the presence of both CD90 and amphiregulin in human chronic pancreatitis by studying in histology sections of pancreas tissue. The demonstration of double positive cells underlines the flow cytometry data and suggests the involvement of ILC2 in pancreatic remodeling also in humans (Fig. 8h).

### Treg cells promote acinar cell regeneration and help to prevent loss of exocrine function
We next analyzed whether Treg-dependent expression of growth factors also affects acinar cell regeneration and the exocrine function of the pancreas in CP. Labeling of Ki67 and α-amylase in pancreatic sections of DT-treated mice showed a significant reduction in the number of proliferating acinar cells, compared to PBS-treated controls. This results in a loss of exocrine tissue (Fig. 9a, b). RT-qPCR analysis detected significantly reduced mRNA levels for genes encoding secretory proteins like α-amylase and T7-trypsinogen (Mus musculus RIKEN cDNA 2210010C04 gene) in pancreatic tissue (Fig. 9c). Finally, we observed that fecal elastase activity was decreased in DT-treated mice compared to PBS controls (Fig. 9d). The amount of fecal elastase is used in clinical practice as a diagnostic marker of pancreatic insufficiency[36]. Two weeks after caerulein treatment the body weight curves indicated a steady decline in the DT-treated animals but not in the PBS-treated control mice. This means that Treg cells are required for the preservation and/or regeneration of pancreatic exocrine tissue and that Treg-mediated inhibition of the type 2 inflammation prevents significant weight loss in our CP model (Fig. 9e).

### Treg cells and Th2 cells are elevated in blood and tissue samples of CP patients
Finally, we investigated numbers of Treg cells or Th2 cells in blood samples of patients suffering from CP. We observed increased numbers of circulating activated CD4+ T cells positive for CD25 or CD69 in those patients (Fig. 10a). Furthermore, we detected significantly increased numbers of FOXP3+CD25+ Treg cells as well as GATA3+ Th2 cells in the blood of CP patients compared to healthy blood donors (Fig. 10b, c). Interestingly we also found increased numbers of TBET+ Th1 cells (Fig. 10d). Pancreatic tissue sections from resected CP patients showed large numbers of FOXP3+ CD4+ Treg cells which had invaded the organ (Fig. 10e). These observations are in line with the results obtained in animal experiments and suggest that FOXP3+ Treg cells are involved also in the remodeling of the pancreas of CP patients.

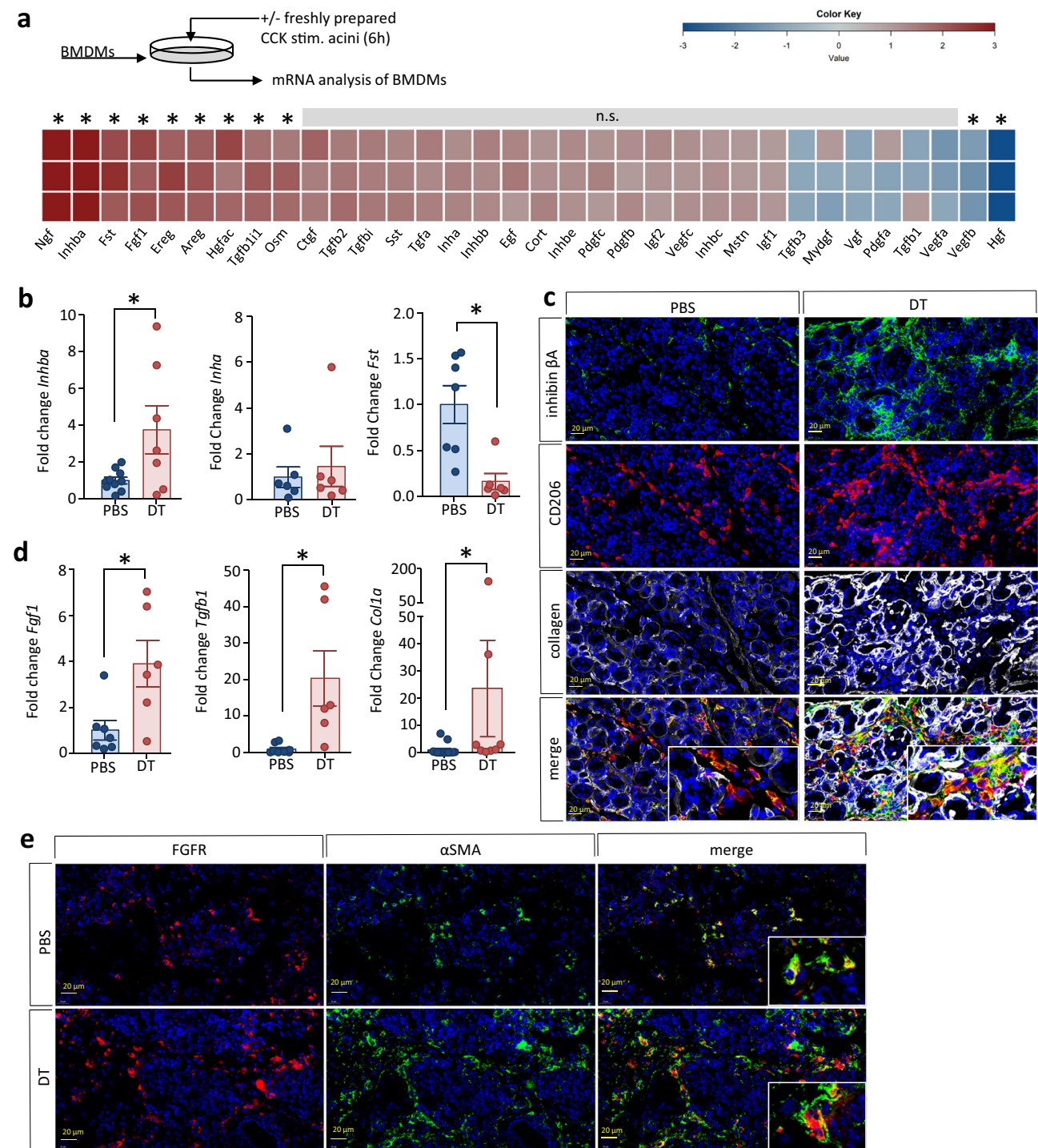

## Discussion

In CP an inflammation-triggered process results in the progressive and irreversible destruction of functional pancreatic tissue and its replacement by fibrotic scar tissue[37]. Till today, a causal therapy or a strategy to limit disease progression are not available. The interaction of pancreatic macrophages and stellate cells is crucial for the progression of fibrosis[13,14,38]. The cytokines IL-4 and IL-13, mainly released by Th2 cells[39] and ILC2s[40], polarize macrophages towards alternatively activated macrophages[13], and link the adaptive to the innate immune response. In the present study we demonstrate a significant change in pancreatic tissue composition after induction of CP in mice, characterized by an infiltration of CCR2+ immune cells, including CD4+ and CD8α+ T cells, macrophages and ILC2s. As a result of this immune reaction, we observed the loss of exocrine and endocrine tissue replaced by fibrosis. Furthermore, we could show that Treg cells were essential to balance the immune reaction and limit the development of fibrosis by suppressing the type 2 immune response driven by alternative activated macrophages, Th2 cells and ILC2s. Alternatively activated macrophage polarization seems to occur in pancreatic tissue as consequence of the migration of Treg cells and Th2 cells in the chronic inflamed organ. Beside the local increase of these cells in the pancreas, we were able to observe a systemic increase of Treg cells and Th2 cells and their released cytokines in mice as well as human samples.

Th2 cells control the polarization of macrophages towards the M2 phenotype via the release of the cytokines IL-4 and IL-13[41]. Previous studies have shown that the IL-4Rα signaling pathway and the presence

**Fig. 7 | Pancreatic macrophages release a panel of growth factors regulating organ fibrosis. a** Bone marrow derived macrophages were isolated from C57Bl/6 mice and co-incubated with CCK-stimulated acini for 6 h. Total RNA was isolated from BMDMs and transcriptome profiling was performed using Affymetrix Gene-Chip Mouse Gene 2.0 ST Arrays. Expression changes of genes encoding growth factors are visualized by a heat map. The gene-specific transcript levels of *Ngf*, *Inhba*, *Fst*, *Fgf1*, *Ereg*, and *Areg* encoding Nerve growth factor, inhibin A subunit β, follistatin, fibroblast growth factor 1, epiregulin, and amphiregulin, respectively, were significantly increased in BMDMs after co-incubation with acini. **b** Growth factor gene expression was also investigated in pancreatic tissue from DEREG mice +/− DT by RT-qPCR. Expression of *Inhba* encoding Inhibin βA was significantly increased in DT-treated mice (*p* = 0.0242, PBS *n* = 10/DT *n* = 7) whereas transcript levels of *Inha* encoding inhibin α were not affected (PBS *n* = 6/DT *n* = 6). Expression of *Fst* encoding follistatin was significantly decreased in DT-treated mice (*p* = 0.0049, PBS *n* = 7/DT *n* = 6). Transcript levels as determined by RT-qPCR were normalized using *RnSs* as internal calibrator gene and were related to the corresponding mRNA amounts in control mice. **c** Immunofluorescence labeling of inhibin βA and CD206 demonstrates co-localisation. **c, d** Gene expression of growth factors FGF1 (*p* = 0.0173, PBS *n* = 7/DT *n* = 6) and TGFB (*p* = 0.0113, PBS *n* = 8/DT *n* = 6) was also upregulated in the pancreas of DT-treated mice and an higher expression of collagen 1 was shown by immunofluorescence labeling and RT-qPCR (two-tailed Mann-Whitney test *p* = 0.0053, PBS *n* = 13/DT *n* = 8). **e** Additional labeling of the FGF-receptor on α-SMA⁺ PSCs demonstrates increased fibrogenesis in DT-treated mice, scale bars represent 20 μm. Significantly differential mRNA levels as detected by microarray-based transcriptome analysis were defined using the following criteria: one-way ANOVA with Benjamini and Hochberg false discovery rate (*p* ≤ .05), signal correction statistics (Ratio Builder software; *p* ≤ .05), and an expression value ratio between the different conditions ≥1.5-fold (*n* = 3). All data were presented as means ± SEM, statistically significant differences were tested by unpaired two-tailed students *t*-test for independent samples, except the collagen 1 gene expression was tested by two-tailed Mann-Whitney test for nonparametric samples, significance levels of *p* < 0.05 are marked by an asterisk (**b**, **d**). Source data are provided as a Source Data file.

of IL-4/IL-13 is crucial for the M2 polarization of CD206⁺ macrophages and determines fibrogenesis during chronic pancreatitis[13]. As source of IL-4, they identify activated PSCs[13], this forms a self-perpetuating cycle in which PSCs drive M2 polarization while CD206⁺ M2 macrophages drive PSC activation via TGF-β. Beside PSCs we could identify pancreatic Th2 cells and ILC2s as source of M2 driving cytokines. The in vivo depletion of IL-4 by neutralizing anti-IL-4 antibody treatment resulted in a significantly reduced Th2 cell differentiation and diminished pancreatic fibrosis. A blockage of the Th2 and ILC2s surface receptor CRTH2 by its antagonist OC000459[31] also results in a diminished infiltration of CD4⁺ T cells in the pancreas in our mouse model of CP. A reduced transcription of IL-4 results in a decreased number of CD206⁺ alternatively activated macrophages and a limited fibrosis and the loss of exocrine tissue. This demonstrates that Th2/ILC2s act as a driving force for organ fibrosis in the pancreas.

Beside Th2 cells, FOXP3⁺CD25⁺ Treg cells are the most prominent cells of the adaptive immune system which were activated local in the pancreas as well as systemic in spleen. A direct or indirect influence of Treg cells on tissue repair, regeneration and fibrosis has been reported for various organs such as skin[42], muscle[43], and lung[44]. Treg cells influence monocyte and neutrophil recruitment, macrophage polarization and T-effector cell differentiation[45] to regulate wound healing and tissue remodeling. In fibrotic lung as well as in liver diseases, FOXP3⁺CD25⁺ Treg cells limit fibrogenesis[44,46]. In our mouse model of CP, as well as in human CP patient's samples, we observed a significant increase of FOXP3⁺ Treg cells and GATA3⁺ Th2 cells in the chronic inflamed pancreas. Our observations are in line with the findings of the group of Habtezion et al, which observe a significant infiltration of Treg cells in pancreatic tissue of CP patients[47,48]. Beside CD4⁺ T cells, also CD8α⁺ T cells could be observed in CP tissue[47,48]. Also CD8α⁺ T cells are regulated by Treg cells[49]. We could show that CD8α⁺ cells are located in chronic pancreatitis tissue of mice and their number is increased in Treg depleted mice. CD8α⁺ cells have an ambiguous effect on fibrogenesis, on the one hand it is known that CD8α⁺ cells can contribute to tissue fibrosis like it is shown for systemic sclerosis[50], on the other hand they can act anti-fibrotic by inducing stellate cell apoptosis like described for fibrotic liver disease[51].

Our experiments demonstrate that during pancreatitis Treg cells suppress the systemic and local type 2 immune response involving cells of the innate as well as the adaptive immune system. The depletion of Treg cells results in a significantly increased systemic and pancreatic Th2/ILC2-response accompanied by increased cytokine levels of IL-4 and IL-10. In the spleen ILCs differentiate into ILC2s and we demonstrate an accumulation of ILC2s in the pancreas. Th2 cells as well as ILC2s are known to release IL-4/IL-13 and promote alternative macrophage polarization[52,53]. In the damaged pancreas we observe a significant shift of macrophage polarization towards alternatively

activated CD206⁺ macrophages which might trigger excessive fibrosis and limit acinar cell regeneration. Alternatively activated macrophages, mainly described as M2-like macrophages, are known to regulate stellate cell activation and fibrosis[13] as well as organ regeneration and acinar cell differentiation[15,54]. The release of various growth factors determine tissue regeneration/fibrosis[55]. Both processes are controlled by growth factors and their counter regulators[18,55]. In vitro analysis demonstrated that macrophages in response to damaged acinar cells react with an increased transcription of various growth factors. Depletion of regulatory T cells permits pancreatic CD206⁺ macrophages to release excessive amounts of various growth factors like TGF-β and FGF known to activate PSCs and to promote fibrosis[19]. Similarly activin, another growth factor of the TGF-superfamily, drives tissue fibrosis in chronic liver[56], kidney[57] and lung[58] diseases and has been reported to activate pancreatic stellate cells and fibroblasts[59]. Inhibin is closely related to activin but has opposite biological effects. Whereas activin, is a homodimer of inhibin βA subunits, inhibin is a heterodimer of inhibin βA with the more distantly related α subunit. Macrophages, in response to damage-associated molecular patterns (DAMP)-signals from damaged acinar cells, upregulate the transcript levels of inhibin βA while those of inhibin α were unchanged, promoting the formation of activin homo-dimers. Activin is known to induce PCSs to secrete extracellular matrix protein synergistically to the effects of TGF-ß[59]. It has also been shown that activin is released from PSCs after TGF-ß stimulation, which may start an autocrine activation loop. Another counter regulator of activin is follistatin, which blocks the interaction of activin with its receptor[60]. Follistatin, which is suppressed in the absence of Treg cells, is known to inhibit the synthesis of collagen and TGF-ß in PSCs and thus terminates the activin autocrine loop[59]. Not only PSCs, but also acinar cells are sensitive to activin stimulation. The presence of activin during acinar cell differentiation suppresses the synthesis of secretory proteins and stops cell proliferation[61]. We assume that in DT-treated animals the observed expression changes of activin and follistatin affect the regeneration of acinar cells, which may explain the enhanced acinar to ductal metaplasia that we see in mice lacking Treg cells. The imbalance between activin on the one hand and inhibin as well as follistatin on the other could contribute to tissue fibrosis and may limit acinar cell regeneration and differentiation. The depletion of Treg cells by DT treatment resulted in an unrestrained Th2/ILC2 response leading to alternative macrophage activation accompanied by a disbalance of growth factors and their counter regulators such as activin/inhibin and follistatin.

As we have shown, Treg cells are able to suppress the macrophage response and growth factor release and prevent uncontrolled tissue fibrosis in CP. Treg cells control pancreatic macrophages via suppression of the Th2 cell differentiation. However, another cell population of the innate immune system, namely the ILCs, could be involved

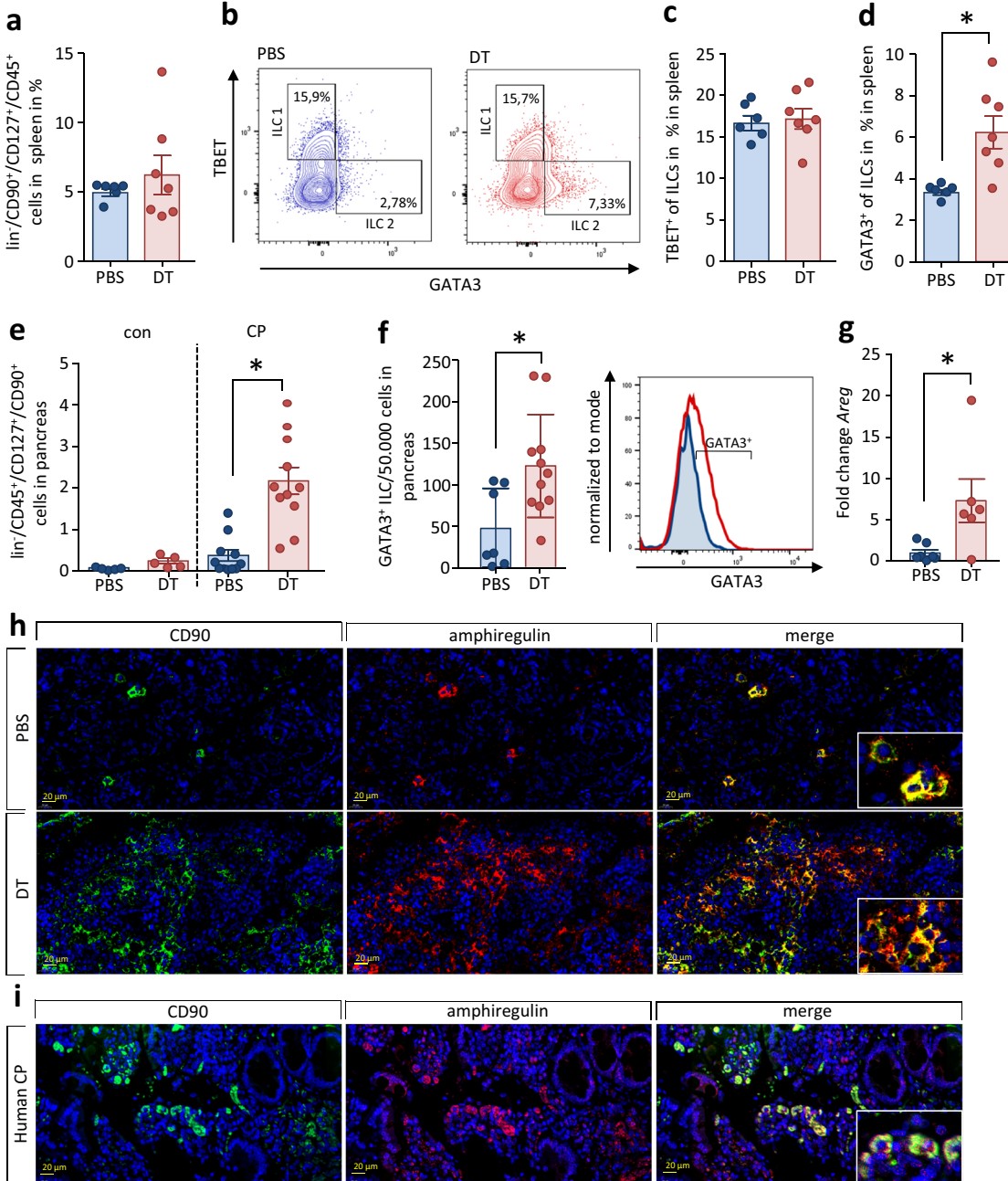

**Fig. 8 | ILC2 are suppressed by regulatory T cells.** Splenocytes of CP animals, additionally treated with DT or PBS were isolated and analyzed by flow cytometry. **a** The fraction of ILCs was discriminated as lin⁻CD45⁺CD127⁺CD90⁺ cells (PBS *n* = 6/DT *n* = 7). **b** The presence of the transcription factors GATA3 and TBET was used to differentiate ILC1 and ILC2. **c, d** Treg depletion increases the number of GATA3⁺ ILC2 (*p* = 0.0064, PBS *n* = 6/DT *n* = 7) but not of TBET⁺ ILC1 (PBS *n* = 6/DT *n* = 7) in spleen. **e** Isolated leukocytes from pancreatic ILCs. The total number of lin⁻CD45⁺CD127⁺CD90⁺ cells was dramatically increased in the absence of Treg cells (*p* < 0.0001, con PBS/DT *n* = 5/ CP PBS/DT *n* = 11), **f** and the number of GATA3⁺ ILC2 was significantly increased (*p* = 0.0155, PBS *n* = 7/DT *n* = 11). **g** Expression analysis of *Areg* encoding amphiregulin in pancreatic tissue by RT-qPCR confirms an increase in Treg-depleted animals during CP (*p* = 0.0253, PBS *n* = 7/DT *n* = 6). Transcript levels as determined by RT-qPCR were normalized using *RnS5* as internal calibrator gene and were related to the corresponding mRNA amounts in control mice. **h, i** Representative immunofluorescence labeling of CD90⁺ AREG⁺ detected AREG in pancreatic ILC2 from CP animals as well as in human tissue sections, scale bars represent 20 μm. All data were presented as means ± SEM, statistically significant differences were tested by unpaired two-tailed students *t*-test for independent samples and significance levels of *p* < 0.05 are marked by an asterisk (**d**–**g**). Source data are provided as a Source Data file.

here. In our model, depletion of Treg cells resulted in a significant accumulation of lin⁻CD45⁺CD127⁺CD90⁺ cells in the pancreas, which we could only detect in a very limited number in CP of wild-type mice. The expression of the transcription factor GATA3 characterizes these cells as ILC2s[62]. Treg cells are able to suppress the proliferation of tissue-resident ILCs via the inducible T cell co-stimulator (ICOS)-ICOS ligand cell contact[63] and in an IL-2-dependent manner[64]. ILC2s in turn can

stimulate the M2-like macrophage response via the release of IL-4/IL-13[65,66]. ILC2s also express the growth factor amphiregulin[67], which was shown to be involved in the TGF-β mediated activation of lung fibrosis and regeneration processes. Amphiregulin is known to activate the epidermal growth factor receptor (EGFR) family signaling pathway[68,69]. Many studies have identified the EGFR-pathway as a key pathway controlling the wound healing response[70–73]. Interestingly increased

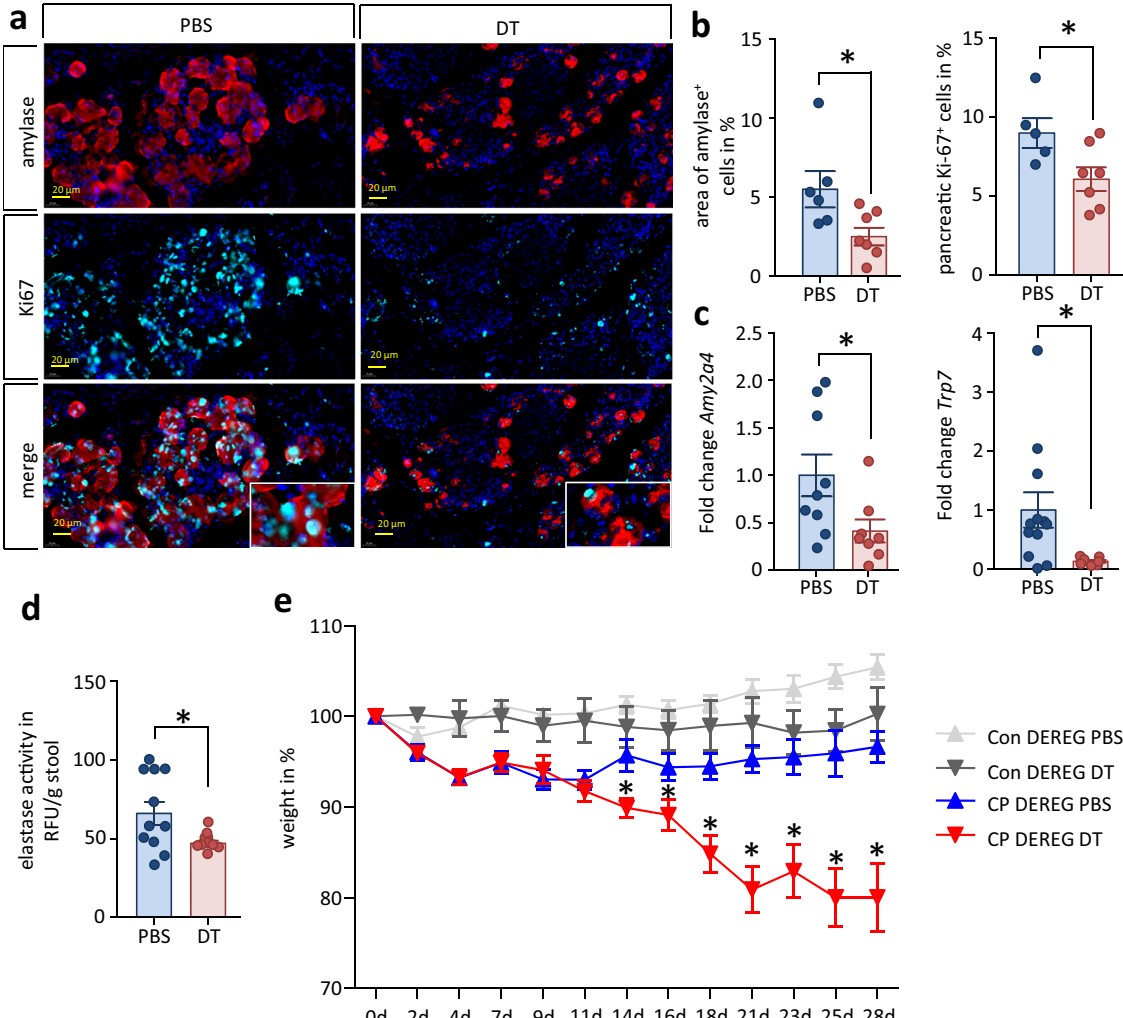

**Fig. 9 | Treg-depletion causes loss of exocrine function. a** Immunofluorescence labeling of α-amylase and Ki-67 indicates less proliferating acinar cells in the Treg depleted group. Scale bars represent 20 μm. **b** Quantification of the immuno-fluorescent labeling confirmed a significantly reduced number of acinar cells (*p* = 0.0314, PBS *n* = 6/DT *n* = 7) and less proliferating Ki67⁺ acinar cells (*p* = 0.0342, PBS *n* = 5/DT *n* = 7) in the pancreas of DT-treated mice. **c** Gene expression of the pancreatic digestive enzymes amylase (*p* = 0.0391, PBS *n* = 9/DT *n* = 8) and trypsi-nogen (*p* = 0.0425, PBS *n* = 12/DT n = 7) as determined by RT-qPCR was significantly decreased during CP and depletion of Treg cells. Transcript levels as determined by RT-qPCR were normalized using *Rn5s* as internal calibrator gene and were related to the corresponding mRNA amounts in control mice. **d** The activity of fecal elastase, a clinical marker of exocrine function, was also significantly decreased in DT-treated mice (*p* = 0.0280, PBS *n* = 11/DT *n* = 10). **e** The decreased exocrine pancreas func-tion is furthermore reflected in body weight loss. Treg depletion in CP animals significantly increases the weight loss over time (one-way ANOVA *p* < 0.0001, Bonferroni posttest, con PBS *n* = 5/DT = 5, CP PBS *n* = 7/DT *n* = 6). All data were presented as means ± SEM, statistically significant differences were tested by unpaired two-tailed students *t*-test for independent samples and one-way ANOVA, Bonferroni posttest, significance levels of *p* < 0.05 are marked by an asterisk. Source data are provided as a Source Data file.

levels of amphiregulin are also found during coronavirus (SARS-CoV) infection, where many survivors develop pulmonary fibrosis. Com-parable to our mouse model of CP, a dysregulated EGFR pathway has also been reported to associate with the development of pulmonary fibrosis-like disease after SARS-CoV infection[74].

In sum our data suggest that Treg cells are essential to prevent uncontrolled fibrogenesis by repressing Th2 cells and ILC2-dependent M2-like macrophage polarization via IL-4/IL-13 release. Depletion of Treg cells initiates a type 2 immune response accompanied by pro-nounced stellate cell activation and extracellular matrix protein pro-duction (Fig. 10f). The irregular release of various growth factors by pancreatic macrophages seems to be responsible for the excessive fibrogenesis. It is interesting to note that despite the increased release of growth factors, damaged acinar cells do not regenerate. Apparently the strong fibrogenic stimulus suppresses the regeneration of the exocrine pancreatic tissue. Data from patients with CP support our observations from animal models, also in patients with CP we could detect systemically as well as locally a significant increased number of Treg cells. For patients suffering from CP a tissue-preserving therapy would be fundamental in order to prevent exocrine and endocrine insufficiency. Our results implicate that Treg cells can limit pancreatic fibrosis while stimulating acinar cell regeneration. Based on our results the Treg/Th2 balance might represent a promising target for the development of a concept for the treatment of CP.

## Methods
### Ethics statement
All animal experiments were approved by the local Animal Care Committee of (LALLF - Landesamt für Landwirtschaft, Lebensmittel-sicherheit und Fischerei Mecklenburg-Vorpommern) and performed in compliance with the ARRIVE guidelines. Studies on human samples were approved by Ethical committee of the university medicine Greifswald. All participants gave written informed consent. Partici-pants were not compensated for their participation in the study.

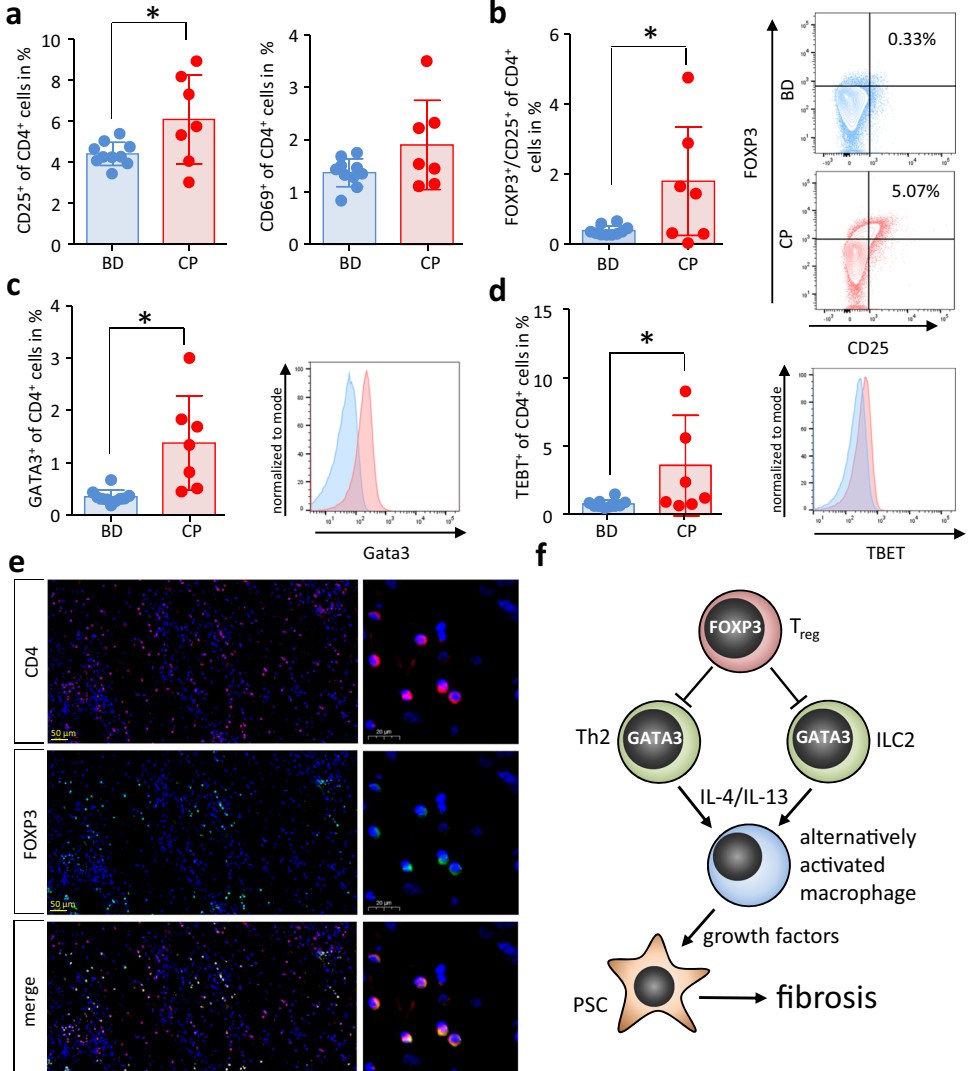

**Fig. 10 | Blood and pancreatic tissue samples of CP patients contain increased numbers of Treg cells.** T cells were isolated from EDTA blood samples of patients (CP) and blood doners (BD) and were analyzed by flow cytometry. **a** The T cell activation marker CD25 ($p = 0.0317$, BD $n = 10$/CP $n = 7$) was significantly elevated in CP patients compared to BD, whereas the elevation of the surface expression of CD69 (BD $n = 10$/CP $n = 7$) did not reach significance. T cell differentiation was analyzed by intracellular transcription factor staining of FOXP3, GATA3 and TBET. **b** FOXP3+CD25+ Treg cells were significantly elevated in the blood of CP patients ($p = 0.0345$, BD $n = 10$/CP $n = 7$), **c** in the same manner as GATA3+ Th2 cells ($p = 0.0026$, BD $n = 10$/CP $n = 7$). **d** Interestingly also TBET+ Th1 cells were significantly increased ($p = 0.0478$, BD $n = 10$/CP $n = 7$). **e** Representative immunofluorescence labeling images of CD4 and FOXP3 illustrate a high number of CD4+FOXP3+ Treg cells in human CP tissue. **f** Summary illustration showing how regulatory T cells control organ fibrosis during chronic pancreatitis by suppressing Th2/ILC2-mediated alternative activation of macrophages via IL-4/IL-13. Alternatively activated macrophages in turn induce activation of pancreatic stellate cells (PSC) which contribute to extracellular matrix production and fibrosis. All data were presented as means ± SEM, statistically significant differences were tested by unpaired two-tailed students t-test for independent samples and significance levels of $p < 0.05$ are marked by an asterisk (**a**–**d**). Source data are provided as a Source Data file.

## Animal model

All mice were kept under pathogen-free conditions in ventilated animal cabinets. All mice were kept in an animal room with a 12-h light-dark cycle at a temperature of 21–24 °C (humidity 50–70%) with access to food and water ad libitum. 8–12 weeks old male or female mice were used for the experiments.

DEREG mice (C57Bl/6 background, bacterial artificial chromosome-transgenic mice expressing a diphtheria toxin (DT) receptor-enhanced green fluorescent protein fusion protein under the control of the *Foxp3* gene locus, allowing selective and efficient depletion of FOXP3+ Treg cells by DT injection) were kindly provided by Jochen Huehn and have been previously described in detail[27]. Wild-type C57Bl/6 mice were purchased from Charles River Laboratories (Sulzfeld, Germany). CP was induced by hourly repetitive caerulein i.p. injection (50 μg/kg/bodyweight) (4030451, Bachem) over 6 hours, three times per week over a time period of four weeks (Supplementary Fig. 1a). Treg cells were depleted three times per week over the period of four weeks of caerulein treatment by 1 μg diphtheria toxin (DT) (D0564, Sigma Aldrich) i.p. injection one hour before the first caerulein dose (Supplementary Fig. 1d). Control animals received phosphate-buffered saline (PBS) instead of DT. This mouse model is well established and results in a successful depletion of Treg cells[25].

To neutralize IL-4 during CP in mice, the Ultra-LEAF purified anti-mouse IL-4 antibody (504138, BioLegend) we injected i.p. 0.75 mg antibody three times a week. Control mice were treated with Ultra-LEAF purified rat IgG1 isotype antibody in the same concentration (Supplementary Fig. 1b).

The CRTH2-specific inhibitor OC000459 (12027, Cayman Chemical) was injected i.p. once daily in a concentration of 30 µg in 1:160 dimethyl sulfoxide (A994.2, Roth). Control mice were treated i.p. with a vehicle (dimethyl sulfoxide) diluted 1:160 in PBS (Supplementary Fig. 1c).

The body weight of the mice was determined over the whole four weeks. After that time all animals were sacrificed. Serum and organs were removed immediately and stored under different conditions for further experiments.

## Antibodies

The following antibodies were used for immunofluorescence labeling and Flow-cytometry analysis: anti-GFP (dilution 1:200, ab6673, abcam), anti-CD3 (dilution 1:200, 100202, BioLegend), anti-CD4 (dilution 1:200, UM800010, origene), anti-Mrc1/CD206 (dilution 1:200, OASA05048, aviva-sysbio), anti-αSMA (dilution 1:200, M0851, Dako), anti-CD90 (dilution 1:200, 14-0900-85, Invitrogen), anti-Amphiregulin (dilution 1:200, sc-5796, SanatCruz), anti-Amphiregulin (dilution 1:200, PA5-27298, Invitrogen), anti-collagen I (dilution 1:200, ab34710, abcam), anti-amylase (dilution 1:200, sc-46657, Santa Cruz), anti-Ki-67 (dilution 1:200, IHC-00375, Bethyl), anti-FGF Receptor (dilution 1:200, 9740S, cell signaling), anti-InhibinβA (dilution 1:200, C9B1223, Bio-Genesis), anti-IL-13 (dilution 1:200, bs-0560R, Bioss Antibodies), anti-IL-33 (dilution 1:200, AF3626, R&D systems), anti-STAT6 (dilution 1:200, ab32520, abcam), anti-GATA3 (dilution 1:200, 14-9966-80, eBioscience), anti-CD25-PE/Cy7 (dilution 1:50, 102016, BioLegend), anti-CD25-PE/Cy7 (dilution 1:50, 302612, BioLegend), anti-CD4-PerCP/Cy5.5 (dilution 1:50, 100433, BioLegend), anti-Cytokeratin 19 (dilution 1:200, ab15463, abcam), anti-CD4-PE (dilution 1:50, 100408, BioLegend), anti-CD4-PerCP/Cy5.5 (dilution 1:50, 100434, BioLegend), anti-CD4-Brilliant-Violet510 (dilution 1:50, 116026, BioLegend), anti-CD4-AlexaFluor488 (dilution 1:50, 317420, BioLegend), anti-CD8α-PE/Cy5 (dilution 1:50, 100710, BioLegend), anti-CD8α-PE (dilution 1:50, 100708, BioLegend), anti-TBET-Brilliant Violet 421 (dilution 1:10, 644815, BioLegend), anti-TBET-PerCP/Cy5.5 (dilution 1:10, 644806, BioLegend), anti-TBET-Brilliant Violet605 (dilution 1:10, 644817, BioLegend), anti-GATA3-PE (dilution 1:10, 653803, BioLegend; dilution 1:10, 130-100-652, MiltenyiBiotec), anti-GATA3-Brilliant Violet421 (dilution 1:10, 653814, BioLegend), anti-RORγt-APC (dilution 1:10, 130-123-840, MiltenyiBiotec), anti-FOXP3-AlexaFluor488 (dilution 1:10, 53-4774-42, Invitrogen), anti-FOXP3-AlexaFluor647 (dilution 1:10, 320114, BioLegend), anti-PTGDR2/CD294-AlexaFlour594 (dilution 1:50, C47774-af594, SAB Signalway Antibody), anti CD69-Brilliant Violet510 (dilution 1:50, 104532, BioLegend), anti-CD69-Brilliant Violet510 (dilution 1:50, 310936, BioLegend), anti-CD11b-PerCP/Cy5.5 (dilution 1:50, 101228, BioLegend), anti-CD206-APC (dilution 1:50, 141708, BioLegend), anti-CD206-PE/Cy7 (dilution 1:50, 141719, BioLegend), anti-CD163-PE (dilution 1:50, 12-1631-82, Invitrogen), anti-CD163-PE/Dazzle (dilution 1:50, 155315, BioLegend), anti-Ly6g-BV421 (dilution 1:50, 127628, BioLegend), anti-lin-AlexaFlour700 (dilution 1:20, 77923, BioLegend), anti-CD45-PerCP (103130, BioLegend), anti-CD45-PE (103106, BioLegend), CD45-PE/Cy5 (dilution 1:50, 103109, BioLegend), anti-CD90-Brilliant Violet605 (dilution 1:50, 105343, BioLegend), anti-CD127-Brilliant Violet650 (dilution 1:50, 135043, BioLegend), anti-GFAP-Alexa Fluor647 (dilution 1:50, 51-9792-82, Invitrogen), anti-CD271-PE (dilution 1:50, 12-9400-42, Invitrogen), anti-CCR2-FITC (dilution 1:50, 150608, BioLegend), anti-IL-4-AlexaFluor488 (dilution 1:50, 504109, BioLegend), anti-IL-13-PE (dilution 1:50, 159403, BioLegend), anti-IFNγ-Brilliant Violet650 (dilution 1:50, 505831, BioLegend), anti-TNF-PE/Cy7 (dilution 1:50, 506324, BioLegend), anti-IL-10-APC (dilution 1:50, 505010, BioLegend) PE/Cyanine7 Rat IgG2b, Isotype Ctrl Antibody (dilution 1:50, 400617, BioLegend), Brilliant Violet 510 Armenian Hamster IgG Isotype Ctrl Antibody (dilution 1:50, 400941, BioLegend), Brilliant Violet 421 Mouse IgG2b, Isotype Ctrl Antibody (dilution 1:50, 400341, BioLegend), Brilliant Violet 421 Mouse IgG1, Isotype Ctrl Antibody (dilution 1:50, 400157, BioLegend), Brilliant Violet 650 Rat IgG1, Isotype Ctrl Antibody (dilution 1:50, 400437, BioLegend), PerCP/Cy5.5 mouse IgG1, Isotype Ctrl Antibody (dilution 1:50, 400150, BioLegend), IgG1 Antibody, anti-mouse APC (dilution 1:50, 130-117-099, MiltenyiBiotec), AlexaFluor488 Rat IgG1, Isotype Ctrl (dilution 1:50, 400417, BioLegend), PE Mouse IgG2b, Isotype Ctrl (dilution 1:50, 400313, BioLegend), PE Mouse IgG1, Isotype Ctrl (dilution 1:50, 551436, BD), APC Rat IgG2b, Isotype Ctrl (dilution 1:50, 400612, BioLegend), anti-goat-Cy3 (dilution 1:200, 705-165-147, Jackson ImmunoResearch), anti-mouse-FITC (dilution 1:200,115-095-146, Jackson ImmunoResearch), anti-mouse-Cy3 (dilution 1:200, 115-165-166, Jackson ImmunoResearch), anti-mouse-Cy5 (dilution 1:200, 115-175-146, Jackson ImmunoResearch), anti-rabbit-FITC (dilution 1:200, 711-095-152, Jackson ImmunoResearch), anti-rabbit-Cy3 (dilution 1:200, 111-165-144, Jackson ImmunoResearch), anti-rat-Cy3 (dilution 1:200, 112-165-062, Jackson ImmunoResearch), AlexaFluor647 donkey anti-rabbit (dilution 1:200, A31573, Invitrogen).

## Flow cytometry analysis

Spleen was homogenized with a 70 µm cell strainer. Splenocytes were washed with PBS and centrifuged at $300 \times g$ for 6 min. The cell pellet was resuspended in 1 mL 1× lysis buffer (10× buffer: 1.5 M $NH_4Cl$, 0.1 M $KHCO_3$, 10 mM EDTA•2Na) and incubated for 5 min at room temperature. The reaction was terminated with PBS.

To study the cytokine production of splenocytes, $2 \times 10^6$ cells per well were stimulated with a cell activation cocktail (423302, BioLegend), Brefeldin A (420601, BioLegend) and Monensin (420701, BioLegend) in a proliferation medium (TexMACS Medium, 130-097-196, MiltenyiBiotec; +10% FCS, +25 mM β-Mercaptoethanol) for 4 h at 37 °C in a $CO_2$ incubator.

Pancreatic tissue was dissociated with the Multi Tissue Dissociation Kit 1 (130-110-201, MiltenyiBiotec). First, the pancreas was transferred to serum-free DMEM and the enzyme mix. The tissue was homogenized with the gentleMACS Dissociator (130-093-235, MiltenyiBiotec) at 120 runs for 37 s. After incubation at 37 °C for 20 min under continuous rotation the samples were again dissociated at 168 runs for 37 s. Acinar cells and extracellular remains were removed by filtration through a 70 µm cell strainer. Subsequently the suspension was centrifuged at $300 \times g$ for 6 min.

Fresh peripheral blood was collected from seven chronic pancreatitis patients and 10 healthy volunteers in BD Vacutainer K2E EDTA tubes (367864, BD). Informed consent was obtained from all persons. 1 mL peripheral blood was used to isolate CD3+ cells (CD3 MicroBeads, 130-050-101, MiltenyiBiotec).

After washing with FACS buffer, $1 \times 10^6$ cells per tube were pre-incubated with 1 µL FcR Blocking Reagent (130-092-575, MiltenyiBiotec) to block non-specific Fc-mediated interactions. Zombie NIR Fixable Viability Kit (423106, BioLegend) was used to exclude dead cells. Next, extracellular markers (1:50 CD4, CD69 and CD25; 1:20 lin-; 1:50 CD90, CD127 and CD45; 1:50 CD11b, Ly6g, CD163, CD206, GFAP, CD271) were labeled by adding the antibody cocktail and incubated at 4 °C for 30 min. After fixation and permeabilization (Transcription Factor Staining Buffer Set, 130-122-981, Miltenyi Biotec; Fixation Buffer, 420801, BioLegend) the cell suspensions were again treated with FcR Blocking Reagent and labeled with the intracellular antibody cocktail (1:10 FOXP3, GATA-3, and TBET; 1:50 IL-4, IL-13, IFNγ, TNF and IL-10). Finally, the samples were analyzed by flow cytometry (BD, *LSRII*) and calculated by *FlowJo* (Supplementary Fig. 3).

## Human sample collection

Chronic pancreatitis EDTA blood samples were collected at the university medicine Greifswald after approval by the Ethical committee of the university medicine Greifswald (III UV 91/03). All patients gave written and informed consent. Control blood was drawn in EDTA from healthy blood donors who gave written and informed consent. This

procedure was approved by the Ethical committee of the university medicine Greifswald (BB 014/14). Human chronic pancreatitis tissue samples were collected in the context of the ChroPac trial (ISRCTN38973832).

## Histology, immunohistochemistry, and immunofluorescence

Pancreas and lymph nodes were removed immediately from the sacrificed mice and fixed in 4.5% formaldehyde for paraffin embedding and for cryo-embedding in TissueTec.

Paraffin-embedded tissue samples were cut in 2 μm slides and afterwards used for Masson Goldner (100485, Merck Millipore) and Azan staining (12079, Morphisto).

Immunofluorescence labeling was performed from 2 μm cryo slides. The antibodies were used in a 1:200 dilution and incubated over night at 4 °C. The appropriate secondary antibody was used also in a 1:200 dilution, 1 h at room temperature.

Human chronic pancreatitis tissue samples was collected in the context of the ChroPac trial (ISRCTN38973832)[75], staining of Areg and CD90 was performed like previously described for mouse tissue samples.

## Serum cytokine measurements

The serum cytokine concentration of IL-6, IL-4, IFNγ, TNF, IL-17A, and IL-10 were measured by Cytometric Bead Array (CBA) Mouse Th1/Th2/Th17 CBA Kit (Mouse inflammation kit) (BD 560485 (552364), BD Bioscience, San Jose, CA, USA).

## Serum collagen measurement

The serum collagen concentration was determined using the Soluble Collagen Assay Kit (ab241015, abcam). The fluorescence (Ex/Em = 340/460 nm) was measured in endpoint mode by a fluorometer (FluorStar Optima, BMG Labtech).

## Serum lipase measurement

The enzyme activities of lipase in the serum was measured using the colorimetric kit Lip (Ref. 11821792216) from Roche/Hitacho.

## Fecal Elastase activity analysis in stool samples

Fecal samples were weighed and resuspend in 500 mmol/L NaCl, 100 mmol/L CaCl$_2$ containing 0.1% Triton X-100 and two times soni-cated. Fecal elastase activity was determined by fluorometric enzyme kinetic over 1 h at 37 °C by the usage of 0.12 mM elastase substrate Suc-AAA-AMC (4006305.0050, Bachem)[12]. Kinetics were measured in 100 mmol/l Tris buffer containing 5 mmol/L CaCl$_2$ at pH 8.0.

## Isolation of BMDMs and acini primary cells

Acini were isolated from mouse (C57Bl/6) pancreas by collagenase digestion (Collagenase of *Clostridium histolyticum* (EC.3.4.24.3) from Serva, lot no. 14007, Heidelberg, Germany) under sterile conditions. The pancreas was digested in medium containing 1 mg collagenase for 2 × 15 min, the acinar cells were carefully separated from the tissue by resuspending several times[6]. Cells were maintained and stimulated in Dulbecco's modified Eagle medium containing 10 mM 4-(2-hydro-xyethyl)−1-piperazine ethansulfonic acid (HEPES), 2% of bovine serum albumin (BSA) and 1% Penstrep. Stimulation of acinar cells was performed with 1 μM CCK (Cholecystokinin (CCK) Fragment 26-33 Amide, non-sulphated, Sigma-Aldrich CAS No: 25679-24-7) for 30 min, afterwards cells were centrifuged for 30 s at 70 g and resuspended in fresh media to wash out residual CCK. BMDMs were isolated from femur and tibia of C57Bl/6 mice under sterile conditions as previously described[10,22]. Bone marrow was flushed out of the bones with sterile PBS and passed through a cell strainer (70 μm). Cells were washed with sterile PBS, counted and maintained in six-well plates in a concentration of 2.5 million cells/well with RPMI medium (1% Penstrep and 10% FCS). Six hours after isolation from the bone marrow medium and non-

attached cells were removed and cells were resuspended in fresh medium containing 20 μg/ml M-CSF. After 5–7 days the cells were used for experiments. BMDMs and acini were co-incubated for 6 h, afterward BMDMs were washed carefully to remove residual acini and total RNA was extracted from cells using TRIzol reagent, followed by column purification and quality control.

## RNA isolation and RT-qPCR analysis

The pancreas was removed and directly snap frozen in liquid nitrogen. Total RNA was extracted from pancreas tissue using TRIzol Reagent (15596026, life technologies). Samples were treated with 500 μL TRIzol and homogenized with a TissueLyser. After the addition of 100 μL Chloroform, samples were vortexed and centrifuged at 22,000 × g, 15 min at 4 °C. Subsequently, the upper of the three resulting phases were transferred to a new tube. RNA was precipitated by adding 250 μL isopropyl alcohol. The sample was incubated for 10 min at RT and centrifuged at 22,000 × g, 10 min, 4 °C. The pellet was washed with 500 μL 75% ethanol and centrifuged at 6500 × g for 10 min at 4 °C. After air-drying the pellet was solved in 100 μL A. dest.

RNA samples (2 μg) were transcribed into complementary DNA (cDNA). The cDNA was synthesized using a standard protocol: 2 μg RNA; 5 μM OligodT primers; 75 ng random primers; 0,5 μM dNTP Mix; 1× First Strand Buffer (18080044, Invitrogen); 10 μM DTT; 40 Units RNasin Ribonuclease Inhibitor (N251B, Promega) and 200 Units M-MLV RT (28025013, invitrogen). The total volume per reaction was 20 μL.

The expression of genes of interest was analyzed by reverse transcription-quantitative PCR (RT-qPCR) using the SYBR-green method. The qPCR amplification was performed in a volume of 5 μL containing 1× SYBR Green PCR Master Mix (4309155, applied biosystems), 300 ng gene-specific oligonucleotide primers (reverse and forward) and a 1:10 dilution of cDNA fragments in two technical replicates. Detects transcript levels were normalized to *RnSs* and to the relative expression in control mice. Quantitative mRNA alterations were determined using the $2^{-\Delta\Delta Ct}$-method.

The following primers were used: *RnSs* forward 5′-GCCCGATCTCGTCTGATCTC-3′ reverse 5′-GCCTACAGCACCCGGT ATTC-3′, *Amy2a4* forward 5′- CAAAATGGTTCTCCCAAGGA-3′ reverse 5′-ACATCTTCTCGCCATTCCAC-3′, *Areg* forward 5′-CTGATCTTTG TCTCTGCCATCA-3′ reverse 5′-AGCCTCCTTCTTTCTTTCTGTT-3′, *Col1a* forward 5′-CAGACTGGCAACCTCAAGAA-3′ reverse 5′-CAAGGGTGCT GTAGGTGAAG-3′, *Fgf1* forward 5′-GATGGCACCGTGGATGGGAC-3′ reverse 5′-AAGCCCTTCGGTGTCCATGG-3′, *Fst* forward 5′-AAAACCT ACCGCAACGAATG-3′ reverse 5′-TTCAGAAGAGGAGGGCTCTG-3′, *Ifng* forward 5′-GGATGCATTCATGAGTATTGC-3′ reverse 5′-CCTTTTCC GCTTCCTGAGG-3′, *Il4* forward 5′-AGATCATCGGCATTTTGAACG-3′ reverse 5′-TTTGGCACATCCATCTCCG-3′, *Il10* forward 5′-TTGAAT TCCCTGGGTGAGAAG-3′ reverse 5′- TCCACTGCCTTGCTCTTATTT-3′, *Il13* forward 5′-CCTCTGACCCTTAAGGAGCTT-3′ reverse 5′-ATGTTGGT CAGGGAATCCAG-3′, *Il1b* forward 5′-GAGGACATGAGCACCTTCTTT-3′ reverse 5′-GCCTGTAGTGCAGTTGTCTAA-3′, *Inha* forward 5′-ATGCAC AGGACCTCTGAACC-3′ reverse 5′-GGATGGCCGGAATACATAAG-3′, *Inhba* forward 5′-GATCATCACCTTTGCCGAGT-3′ reverse 5′-TGGTCCT GGTTCTGTTAGCC-3′, *Mrc1* forward 5′-GGCGAGCATCAAGAGTAAAG A-3′ reverse 5′-CATAGGTCAGTCCCAACCAAA-3′, *Tgfb1* forward 5′-CGA AGCGGACTACTATGCTAAA-3′ reverse 5′-TCCCGAATGTCTGACGTA TTG-3′, *Tnf* forward 5′-GCCTCCCTCTCATCAGTTCTAT-3′ reverse 5′-C ACTTGGTGGTTTGCTACGA-3′, *Trp7* forward 5′-CAACTACCCTTCA CTCCTTCAG-3′ reverse 5′-TGCCTGGGTAAGAACTTGTG-3′, *Chil3* forward 5′-TCCAGAAGCAATCCTGAAGAC-3′ reverse 5′-GTCCTTAGCCC AACTGGTATAG-3′, *Acta2* forward 5′-GCCAGTCGCTGTCAGGAACCC-3′ reverse 5′-CCAGCGAAGCCGGCCTTACA-3′.

## Transcriptome analysis of BMDMs

Individual RNA samples were analyzed using Affymetrix GeneChip Mouse Gene 2.0 ST Arrays (Cat. 902118, Thermo Fisher Scientific Inc.,

Waltham, MA) and GeneChip WT PLUS Reagent Kit (Cat. 902280, Thermo Fisher Scientific Inc., Waltham, MA) according to the manufacturer's instructions. Microarray data analysis was performed using the Rosetta Resolver software system (Rosetta Bio Software, Seattle, WA). Significantly different mRNA levels were defined using the following criteria: one-way ANOVA with Benjamini and Hochberg FDR ($p \leq 0.05$), signal correction statistics (Ratio Builder software, Rosetta Resolver) ($p \leq 0.05$), and an expression value ratio between the different conditions of 1.5-fold[10,22].

## Software

Flow cytometric data were analyzed using BD FACS Diva and FlowJo. GraphPad Prism and SigmaPlot were used to present the data and for statistical analysis. The heatmap to visualize the transcriptome array data was created with RStudio.

## Reporting summary

Further information on research design is available in the Nature Research Reporting Summary linked to this article.

## Data availability

Microarray data have been deposited in the National Center for Biotechnology Information (NCBI) Gene Expression Omnibus (GEO) database and are accessible through the following GEO accession number: GSE192517. All data supporting the findings of this study are available within the paper and its Supplementary Information files. Source data are provided with this paper.

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

## Acknowledgements
The authors would like to thank Jochen Huehn for providing the DEREG mice. We also thank Kathrin Gladrow, Diana Krüger, Susanne Wiche and Jenny Radel for their technical support. This work was supported by: Deutsche Forschungsgemeinschaft DFG SE 2702/2-1—M.S., Deutsche Forschungsgemeinschaft DFG SE 2702/2-3—M.S., Deutsche Forschungsgemeinschaft DFG AG 203/4-1—A.A.A., Deutsche Forschungsgemeinschaft DFG GRK 1947—M.S., Deutsche Forschungsgemeinschaft DFG GRK 2719—M.S., A.A.A., B.M.B., and U.V. Deutsche Forschungsgemeinschaft DFG MA 4115/1-2/3—J.M., Deutsche Forschungsgemeinschaft DFG SFB1321: Project-ID 329628492— J.M., the PePPP center of excellence MV (ESF/14-BM-A55-0045/16)— F.U.W., M.M.L., and the EnErGie/P2 Project (ESF/14-BM-A55-0008/18)—A.A.A. and M.M.L.

## Author contributions
Concept of the study M.S., F.U.W., and Juliane Glaubitz. Data acquisition and interpretation: Juliane Glaubitz, A.W., A.A.A., M.S., Janine Golchert, G.H., U.V., B.M.B., and T.T. Writing committee M.S., F.U.W., Juliane Glaubitz, J.M., and M.M.L. Correction of manuscript and approval of final version: all.

## Funding

## Competing interests
The authors declare no competing interests.
