## [Peer Review File · Nature Communications]

In chronic pancreatitis CD25+/FOXP3+ regulatory T cells control pancreatic fibrosis by suppression of the type 2 immune responseREVIEWER COMMENTS

Reviewer #1 (Remarks to the Author):

The article by Glaubitz et al presents the role of Treg cells in suppressing the type 2 immune responses to attenuate fibrotic tissue remodeling during chronic pancreatitis. The findings of this study suggest that the depletion of Treg cells results in a significant increase in collagen production in a mouse model of repetitive caerulein-induced chronic pancreatitis (CP). In particular, the data presented suggest that Treg cells elevated in CP function as a positive regulator of macrophage activation and fibrotic remodeling in the pancreas. Overall, the data presented in this manuscript points to an interesting mechanism of regulation of fibrosis in the pancreas by Treg. However, the possible Treg cell regulation of Th2 responses and the pro-fibrotic effects of elevated Th2 responses are not new in the field. Also, the data presented are superficial to suggest a possible association but lacks in-depth molecular mechanisms. Some of the key weaknesses are highlighted below.

Major Comments:

- 1. The data presented in figure 1 suggest that increases in Treg cells and Th2 responses are associated with fibrosis in CP. However, the data presented is not convincing. In particular, the use of random images IF staining to demonstrate an increase in collagen or Treg cells is not rigorous. The use of biochemical methods is needed to demonstrate an increase in collagen and also quantification of T cell subsets using quantitative 3D imaging methods in the pancreas during CP compared to controls is needed. Also, it is important to use appropriate isotype staining controls to demonstrate the specificity of immunofluorescence and FACS staining.**
- 2. The lack of evidence on Treg cell accumulation in pancreatitis patients is a weakness. Do the patients with chronic pancreatitis have elevated Treg and Th2 cells in their circulation and biopsies?**
- 3. It is concerning to see no significant change in Th1 cells in spleens of CP mice despite an increase in Th2 responses. Do the percentage of Th1 cells in the pancreas and systemic Th1 cytokine levels in the CP model remain unchanged?**
- 4. The data in figure 2 suggests that Th2 cytokines are elevated with DT treatment in CP mice. It is important to establish the source of Th2 cytokine in CP mice with the deletion of Treg cells as multiple hematopoietic and non-hematopoietic cells are implicated in producing Th2 cytokines.**
- 5. Mouse model of CP is a heterogeneous and complex disease model. It is not established why Th2 polarized macrophages are considered as a downstream modifier of pathology in Treg depleted mice. Also, there is no data to demonstrate an observed increase in M2 macrophages is due to elevated Th2 T cells with the depletion of Treg cells.**
- 6. Considering macrophage heterogeneity and their activation status in CP, the findings using BMDM in co-culture studies may not be relevant and descriptive.**

Reviewer #2 (Remarks to the Author):

The study by Glaubitz identified Tregs as central regulators of the fibroinflammatory via suppression of the type 2 immune response during chronic pancreatitis. The authors used genetic mouse model to selectively deplete FoxP3-positive cells (known as Tregs)

to study its central role in chronic pancreatitis and the underlying mechanism by which Tregs modulate pancreatic fibrosis and organ remodelling. Overall, this is an interesting and very important work, with the potential to better understand the pathophysiology of chronic pancreatitis and to identify novel therapeutic target for pancreatic fibrosis. However, it would be really helpful to provide a detailed explanation about how Tregs potentially modulate various immune cell types during chronic pancreatitis and how the altered immune cell profiles in the pancreas, not just in spleen can contribute to the development and progression of chronic pancreatitis. I also have several specific comments below:

1. Most evaluation of the immune cells were performed in the spleen, unless solid evidence or references were introduced at the beginning of the study, it would be really important to also assess the changes of immune cells in the pancreas.
2. Since Tregs are known to be immunosuppressive and the balance of immune system may play crucial role in the physiological and pathophysiological condition. It would be of great importance to show the changes of T cell subtypes, such as CD4+ (Th1, Th2, and Th17), CD8+ etc in the spleen and possibly locally within the pancreas.
3. Most of the results did not show important controls, specifically DEREK mice that were treated with PBS and DT without inducing chronic pancreatitis. Since it was reported Treg deficiency causes lethal and CD4+ T cell-driven autoimmune disease (PMID: 27994068).
4. Also, it would be really helpful to provide the timeline for DT treatment related to the induction of chronic pancreatitis. Is there any organ-specific inflammation in those Tregs depleted mice before induction of chronic pancreatitis? What about depletion of Tregs about inducing chronic pancreatitis?
5. It is unclear the direct link between suppression of the splenic type 2 immune response and the progression of chronic pancreatitis. What happened after Treg depletion globally and locally during chronic pancreatitis?
6. It is unclear through which mechanism that Tregs modulates CD206+ macrophages or macrophage polarization? Why the authors specifically assessed CD206+ macrophages? What about other immune cells in the pancreas?
7. Since the focus of this study is the role of Tregs, it is difficult to understand why the author selected the growth factors from co-culture model of BMDM and CCK-stimulated acini. It would be really helpful to provide some evidence.
8. Assessing the changes of ILC2 in the pancreas is interesting, however, it is would be important to demonstrate and explain how Tregs potentially modulates ILC2.
9. The resolution of the images needs to be improved in Figure 1a and g, Figure 3a and f, Figure 4c and e, Figure 5h and i, and Figure 6a.
10. Supplement Fig 1 is not for ILCs. Please check.

Reviewer #3 (Remarks to the Author):

Glaubitz et al explore immune control of chronic pancreatitis using a model induced by caerulein. The authors continue work published in a report last year. They document increased type 2 responses including Th2 cells and M2 macrophages. They then use the DEREK model to demonstrate Treg-dependent control of Th2 responses, ILC2 expansion, fibrosis, and tissue destruction. Overall, the experiments are performed well and fairly convincing. However, there are few points to address. Moreover, the studies as shown document many correlations with Treg depletion, but lack clear mechanistic insight. This is discussed in specific comments.

1. There is a concern about some of the flow cytometry analysis. In Figure 1a, the immunofluorescence data suggests a dramatic increase in CD206+ macs. However, the flow cytometry shows a very modest difference and the CD206+ population is not even a clearly delineated population. Is there a possibility that the enzyme dissociation of the tissue is destroying the CD206 epitope? Could the IF be quantitated to provide this important piece of information?

2. Similar to the point above, the GATA3 staining in Fig. 1 shows a shift in the entire population. The Th2 population should be confirmed with another parameter; either IL-4 and/or IL13 expression or ST2 staining.

3. There are many correlations following Treg depletion (increased Th2, increased M2, increased ILC2, altered cytokine profiles, increased fibrosis and tissue destruction, weight loss). However, there are no experiments functionally linking these observations to the pancreatitis that directly support the conclusion(s) in the title and abstract. The simplest would be to treat mice with anti-IL-4 to show that IL-4 is critical for the effects of Treg depletion. This would also determine if the changes observed in Fig. 4 and 6, which are less associated with IL-4 biology, are due to IL-4 or another pathway. On the point, the authors should be more careful about making causal conclusions with only correlative data.

4. The CD90 IF in Fig. 5h and I is problematic because CD90 is not restricted to ILCs. It is also expressed on T cells. The authors need to distinguish these populations.

5. Since the authors have access to patient samples (Fig. 5i), have they stained for CD4/Foxp3 to show that Tregs are present in CP patient biopsies? This would seem to be central to the report's conclusions that Tregs limit the CP inflammation and disease.

REVIEWER COMMENTS

Reviewer #1 (Remarks to the Author):

The article by Glaubitz et al presents the role of Treg cells in suppressing the type 2 immune responses to attenuate fibrotic tissue remodeling during chronic pancreatitis. The findings of this study suggest that the depletion of Treg cells results in a significant increase in collagen production in a mouse model of repetitive caerulein-induced chronic pancreatitis (CP). In particular, the data presented suggest that Treg cells elevated in CP function as a positive regulator of macrophage activation and fibrotic remodeling in the pancreas. Overall, the data presented in this manuscript points to an interesting mechanism of regulation of fibrosis in the pancreas by Treg. However, the possible Treg cell regulation of Th2 responses and the pro-fibrotic effects of elevated Th2 responses are not new in the field. Also, the data presented are superficial to suggest a possible association but lacks in-depth molecular mechanisms. Some of the key weaknesses are highlighted below.

Major Comments:

1. The data presented in figure 1 suggest that increases in Treg cells and Th2 responses are associated with fibrosis in CP. However, the data presented is not convincing. In particular, the use of random images IF staining to demonstrate an increase in collagen or Treg cells is not rigorous. The use of biochemical methods is needed to demonstrate an increase in collagen and also quantification of T cell subsets using quantitative 3D imaging methods in the pancreas during CP compared to controls is needed. Also, it is important to use appropriate isotype staining controls to demonstrate the specificity of immunofluorescence and FACS staining.

Response: Thank you for the constructive suggestion to quantify the T-cell subclasses in the pancreas. However, instead of using quantitative 3D imaging methods, we isolated leukocytes from the pancreas of mice and analysed by flow cytometry their altered differentiation (Th1/Th2/Th17 and Treg) in presence of CP. We detected a significant number of CD4⁺ T cells which migrated into the pancreas of CP animals. Analysis of the transcription factors Foxp3, Gata3, Tbet and Ror γ t showed that Foxp3⁺ Tregs represent the most frequent population of CD4⁺ T-helper cells in the chronic inflamed pancreas. Gata3⁺ Th2 cells could also be detected as well as small numbers of Tbet⁺ Th1-cells. In contrast Ror γ t⁺ Th17 cells were hardly detectable (**new Fig.2 a-d**). These new data confirmed a Treg/Th2 differentiation not only in the spleen, but also in pancreatic tissue of CP mice. Labeling

experiments of Foxp3/CD4 Tregs or Stat6/CD4 Th2-cells in human chronic pancreatitis tissue are in line with the data from animal experiments (**new Fig. 10e and f**). The newly obtained data were integrated into the revised manuscript as (**Fig. 2 and Fig 10e-f**).

IF random pictures can provide an overview of the histologic changes, but truly are not a rigorous evaluation method. Therefore we scanned whole slides of histological tissue sections (not individual images) and quantified them entirely using the Cell Quant software as shown below. Counting of cell populations was performed by analysis of the whole slide after excluding non-pancreatic tissue like lymph nodes or fatty tissue. Cell counts are presented as a percentage of the total number of cell nuclei in the whole tissue section (for example **new Fig.2 a and b**). Results from the histologic quantification complement the flow cytometric analysis data and support our previous findings.

Figure for review only: Immunofluorescent labeling of CD206 in CP tissue was scanned and quantified by the usage of cell quant software (sysmex) (a). First we define the area for quantification to exclude artefacts or non-pancreatic tissue like lymph nodes, than we use a fluorescent quantification scenario to quantify CD206⁺ cells, negative cells are shown in blue in the overlay, weak positive cells in yellow, medium positive cells in orange and strong positive cells in red. Finally, we calculate all positive cells against all negative cells to quantify cell count in immunofluorescent staining. To determine tissue remodeling we used pattern quant software from Sysmex (d). Whole slides were scanned, pancreatic tissue was marked for quantification and tissue structures and staining color was used to define tissue for quantification by the software (exocrine and endocrine tissue red, fibrosis green and free space yellow). Finally, the measured area was quantified in % of total measurement area.

We were able to confirm organ fibrosis by quantifying Masson Goldner staining. As suggested by the reviewer, we also determined a significant increase of soluble serum collagen in CP animals using an Soluble Collagen Assay Kit. (**new Fig. 3I**). The quantification of collagen in tissue by biochemical methods is difficult because collagen can only be detected in native samples, but the pancreas expresses a variety of proteases that degrade collagen under native conditions. Therefore, detection by immunohistochemistry or immunofluorescence labeling is the most promising method to demonstrate collagen accumulation. On the other hand, biochemical staining of fibrosis via Azan blue or Masson Goldner is independent of antibodies and another powerful method to quantify tissue

fibrosis. The results from histology and serum collagen measurements were included into the revised manuscript (**new Fig. 3 g-l**).

Concerning the reviewers request to assess the specificity of the used isotype antibodies we performed flow cytometry analysis using isotype antibodies, to exclude false positive signals and determine the gating strategy. All Isotype antibodies we used were included in the material and methods section of the revised manuscript. “, *Brilliant Violet 510™ Armenian Hamster IgG Isotype Ctrl Antibody (400941, BioLegend)*, *Brilliant Violet 421™ Mouse IgG2b, Isotype Ctrl Antibody (400341, BioLegend)*, , *Brilliant Violet 650™ Rat IgG1, Isotype Ctrl Antibody (400437, BioLegend)*, *PerCP/Cy5.5 mouse IgG1, Isotype Ctrl Antibody (400150, BioLegend)*”, *IgG1 Antibody, anti-mouse APC (130-117-099, MiltenyiBiotec)*, *PE/Cyanine7 Rat IgG1, Isotype Ctrl Antibody (400416, BioLegend)*, *AlexaFluor488 Rat IgG1, Isotype Ctrl (400417, BioLegend)*, *PE Mouse IgG1, Isotype Ctrl (551436, BD)*, *APC Rat IgG2b, Isotype Ctrl (400612, BioLegend)*.

Figure for reviewer only: Isotype antibodies were used for flow cytometry analysis to verify the specificity of staining (black is isotype antibody, red line is the marker specific antibody).

The specificity of labeling was tested against positive controls like spleen or lymph nodes, as well as negative controls like untreated mice pancreas which showed no positive signal of infiltrating cells. Taking together flow cytometry analysis of isolated cells from the pancreas and staining were performed in independent samples and lead to the same result. Also the usage of different antibodies for the same target protein on the one hand for flow cytometry and on the other hand for immunofluorescent labeling with the same result suggests the specificity of the antibodies.

2. The lack of evidence on Treg cell accumulation in pancreatitis patients is a weakness. Do the patients with chronic pancreatitis have elevated Treg and Th2 cells in their circulation and biopsies?

Response: We thank the reviewer for raising this important point.. As we had access to paraffin embedded pancreas samples of resected chronic pancreatitis patients from the ChroPac trial (ISRCTN38973832) [1], we were able to analyse some tissue sections and observed a high amount of Foxp3⁺/CD4⁺ T-cells within the chronic pancreatitis tissue (**new Fig. 10e**). Additionally, we collected blood samples from patients with CP and healthy blood donors for flow cytometry analysis of

circulating T-cells and observed an increase of Foxp3⁺/CD25⁺ Tregs and Gata3⁺ Th2-cells in patients with CP (**new Fig. 10a-d**). Increased circulating Treg and Th2-cells as well as a Treg and Th2 infiltrates in the chronic inflamed tissue support our observation in the mouse model of CP. The new data were included in the revised manuscript (**Fig. 10**).

1. Diener MK, Hüttner FJ, Kieser M, Knebel P, Dörr-Harim C, Distler M, Grützmann R, Wittel UA, Schirren R, Hau HM, Kleespies A, Heidecke CD, Tomazic A, Halloran CM, Wilhelm TJ, Bahra M, Beckurts T, Börner T, Glanemann M, Steger U, Treitschke F, Staib L, Thelen K, Bruckner T, Mihaljevic AL, Werner J, Ulrich A, Hackert T, Büchler MW; ChroPac Trial Group. Partial pancreatoduodenectomy versus duodenum-preserving pancreatic head resection in chronic pancreatitis: the multicentre, randomised, controlled, double-blind ChroPac trial. *Lancet*. 2017 Sep 9;390(10099):1027-1037.

3. It is concerning to see no significant change in Th1 cells in spleens of CP mice despite an increase in Th2 responses. Do the percentage of Th1 cells in the pancreas and systemic Th1 cytokine levels in the CP model remain unchanged?

Response: In order to analyse the Th1 response T cells of the spleen as well as the T cells infiltrating the pancreas were examined more closely. As previously shown, we found no significant increase in Th1 cells in the spleen using the transcription factor Tbet (**Fig. 1b**). However, in the pancreas we were able to detect Tbet⁺ Th1 cells (about 2.1% of total CD4⁺ cells). This is significantly less compared to Gata3⁺ Th2 cells (~4.9%) and Foxp3⁺ Tregs (~8.7%) (**new Fig. 2d**). Our findings are comparable to the observations of Lee B et al. in human hereditary pancreatitis patients which show a heterogeneous infiltration of T-cells in the chronic inflamed pancreas [1]. In contrast to their observations, in our mouse model Foxp3⁺ Tregs represented the most prominent population of CD4⁺ T-cells in the organ. Next we analysed the cytokine production of CD4⁺ T-cells isolated from spleen. We analysed the intracellular expression of Th1 cell marker TNF α and IFN γ , as well as Th2 cell marker IL4 and IL13. IL4 and IL13 producing cell numbers were increased and a slight increase of IFN γ producing T-cells could also be measured, whereas TNF α ⁺ T-cells remained unaffected. Serum cytokine levels showed no significant increase for the Th1 cytokines IFN γ and TNF α whereas IL4 was elevated 4 weeks after induction of CP. Quantitative RT-PCR analysis of the gene expression of *IL4*, *IL13*, *IFN γ* , *TNF α* and *IL10* confirmed these findings (**new Fig. 1g-i**). The newly obtained data were included in the revised manuscript (**Fig. 1g-I and Fig. 2d**). In human CP blood samples we also analysed and observed an increased amount of circulating Tbet⁺ Th1 cells (**Fig. 10d**). This result was also included in the revised manuscript.

1. Lee B, Adamska JZ, Namkoong H, Bellin MD, Wilhelm J, Szot GL, Louis DM, Davis MM, Pandol SJ, Habtezion A. Distinct immune characteristics distinguish hereditary and idiopathic chronic pancreatitis *J Clin Invest*. 2020 May 1;130(5):2705-2711.

4. The data in figure 2 suggests that Th2 cytokines are elevated with DT treatment in CP mice. It is important to establish the source of Th2 cytokine in CP mice with the deletion of Treg cells as multiple hematopoietic and non-hematopoietic cells are implicated in producing Th2 cytokines.

Response: Via flow cytometry analysis of *in vitro cell* activation cultures of splenocytes from CP and control mice we identified splenic CD4⁺ T-cells as the source of the Th2 cytokines IL4 and IL13 (**new Fig. 1h**). Immunofluorescence co-labeling of IL13 in CD4⁺ T-cells in CP-tissue showed an increase of IL13 producing CD4⁺T-cells in the pancreas (**new Fig. 2f**). In previous experiments we have shown that macrophages by themselves are not the source of IL4/IL13 [1]. Xue et al. have reported that pancreatic stellate cells can release IL4/IL13 in crosstalk with CD206⁺ M2 macrophages [2].

1. Sandler M, van den Brandt C, Glaubitz J, Wilden A, Golchert J, Weiss FU, Homuth G, De Freitas Chama LL, Mishra N, Mahajan UM, Bossaller L, Völker U, Bröker BM, Mayerle J, Lerch MM. NLRP3 Inflammasome Regulates Development of Systemic Inflammatory Response and Compensatory Anti-Inflammatory Response Syndromes in Mice With Acute Pancreatitis. *Gastroenterology*. 2020 Jan;158(1):253-269.e14.
2. Xue J, Sharma V, Hsieh MH, Chawla A, Murali R, Pandol SJ, Habtezion A. Alternatively activated macrophages promote pancreatic fibrosis in chronic pancreatitis *Nat Commun*. 2015 May 18;6:7158.

5. Mouse model of CP is a heterogeneous and complex disease model. It is not established why Th2 polarized macrophages are considered as a downstream modifier of pathology in Treg depleted mice. Also, there is no data to demonstrate an observed increase in M2 macrophages is due to elevated Th2 T cells with the depletion of Treg cells.

Response: To investigate this important question, we characterised the inflammatory infiltrate of the pancreas in more detail. We observed a prominent infiltration of CD4⁺ T-cells, and the analysis of T-cell differentiation showed that the major sub-populations of CD4⁺ T-cells which infiltrate the chronic inflamed pancreas of mice consisted of Foxp3⁺ regulatory T-cells and Gata3⁺ Th2 cells (**new Fig. 2a-d**). Additionally, we could demonstrate that CD4⁺ Th2-cells are the source of the cytokines IL4 and IL13 (**new Fig. 1h**). Immunofluorescence co-labeling of CD4 and IL13 in CP tissue gave evidence that T-cells secrete IL13 in the pancreas (**new Fig. 2f**). IL4 and IL13 are the major cytokines which induce M2 polarization of macrophages [1]. To prove that Th2-cells are involved in alternative macrophage activation during CP, we blocked Th2 cells “*in vivo*” by using the CRTH2 antagonist OC000459. OC000459 inhibits the activation of Th2 cells [2] and their recruitment into the diseased organ. In C57Bl/6 mice, CP induction was complemented with OC000459 treatment and these animals showed a lower infiltration of CD4⁺ T-cells and of CD4⁺/Stat6⁺ Th2 cells into the pancreas. Furthermore, the number of CD206⁺ macrophages in the pancreas was significantly reduced, which also resulted in a reduction of α SMA⁺ pancreatic stellate cells. Histological staining revealed that tissue fibrosis was reduced while the amount of α -amylase expressing exocrine tissue was increased compared to animals treated with vehicle only. Finally, gene expression analysis of pancreatic tissue by RT-qPCR demonstrated significantly decreased transcript levels for *Mrc1*, *Col1a*, *Il4*, *Il10*, *Areg* and *Tgfb*, indicating a reduced type 2 immune response in OC000459 treated mice (**new Fig. 4a-j**).

In Addition, we saw a significant increase of CD4⁺ cells in the pancreas of Treg depleted mice (**new Fig. 6 a, b and d**), which suggests that Tregs suppress T-cell migration into the inflamed pancreas. Via IL4 and IL13 release (**Fig. 1h and Fig. 2f**) these Th2-cells induce M2 macrophage polarisation (marked by CD206 expression), which ultimately triggers tissue fibrosis. All new data are included in the revised manuscript.

1. Van Dyken SJ, Locksley RM. Interleukin-4- and interleukin-13-mediated alternatively activated macrophages: roles in homeostasis and disease *Annu Rev Immunol.* 2013;31:317-43.
2. Pettipher R, Vinall SL, Xue L, Speight G, Townsend ER, Gazi L, Whelan CJ, Armer RE, Payton MA, Hunter MG. Pharmacologic profile of OC000459, a potent, selective, and orally active D prostanoid receptor 2 antagonist that inhibits mast cell-dependent activation of T helper 2 lymphocytes and eosinophils *J Pharmacol Exp Ther.* 2012 Feb;340(2):473-82.

6. Considering macrophage heterogeneity and their activation status in CP, the findings using BMDM in co-culture studies may not be relevant and descriptive.

Response: Our data show significant expression changes of several growth factors in the DT-treated DREG mice. In order to identify the source of these growth factors, we performed in addition to immunofluorescence labelling studies co-incubation experiments of isolated macrophages which we co-incubated with acinar cells that previously had been activated to an “in vitro state of pancreatitis” by CCK-stimulation. This experiment aimed for the identification of those factors which are released by macrophages under "in vitro" conditions of pancreatitis. It is well established in the current literature that M2 polarised macrophages strongly influence organ remodelling in the course of CP and are crucial for the replacement of functional exocrine and endocrine pancreatic tissue by fibrosis [1, 2]. Our results show that macrophages release growth factors which are known regulators of organ fibrosis. In particular, TGFβ-mediated activation of pancreatic stellate cells and associated production of extracellular matrix proteins is well studied [3-5]. Our results show that pancreatic macrophages not only release TGFβ, but also secrete a variety of other growth factors or inhibitors such as Amphiregulin and Follistatin, which are involved in the restructuring of the organ during chronic pancreatitis.

1. Wu J, Zhang L, Shi J, He R, Yang W, Habtezion A, Niu N, Lu P, Xue J. Macrophage phenotypic switch orchestrates the inflammation and repair/regeneration following acute pancreatitis injury *EBioMedicine.* 2020 Aug;58:102920.
2. Criscimanna A, Coudriet GM, Gittes GK, Piganelli JD, Esni F. Activated macrophages create lineage-specific microenvironments for pancreatic acinar- and β-cell regeneration in mice *Gastroenterology.* 2014 Nov;147(5):1106-18.e11.
3. Xue J, Sharma V, Hsieh MH, Chawla A, Murali R, Pandol SJ, Habtezion A. Alternatively activated macrophages promote pancreatic fibrosis in chronic pancreatitis *Nat Commun.* 2015 May 18;6:7158.
4. Apte MV, Haber PS, Darby SJ, Rodgers SC, McCaughan GW, Korsten MA, Pirola RC, Wilson JS. Pancreatic stellate cells are activated by proinflammatory cytokines: implications for pancreatic fibrogenesis *Gut.* 1999 Apr;44(4):534-41. doi: 10.1136/gut.44.4.534.

5. Phillips PA, McCarroll JA, Park S, Wu MJ, Pirola R, Korsten M, Wilson JS, Apte MV. Rat pancreatic stellate cells secrete matrix metalloproteinases: implications for extracellular matrix turnover Gut. 2003 Feb;52(2):275-82. doi: 10.1136/gut.52.2.275.

Reviewer #2 (Remarks to the Author):

The study by Glaubitz identified Tregs as central regulators of the fibroinflammatory via suppression of the type 2 immune response during chronic pancreatitis. The authors used genetic mouse model to selectively deplete FoxP3-positive cells (known as Tregs) to study its central role in chronic pancreatitis and the underlying mechanism by which Tregs modulate pancreatic fibrosis and organ remodelling. Overall, this is an interesting and very important work, with the potential to better understand the pathophysiology of chronic pancreatitis and to identify novel therapeutic target for pancreatic fibrosis. However, it would be really helpful to provide a detailed explanation about how Tregs potentially modulate various immune cell types during chronic pancreatitis and how the altered immune cell profiles in the pancreas, not just in spleen can contribute to the development and progression of chronic pancreatitis. I also have several specific comments below:

1. Most evaluation of the immune cells were performed in the spleen, unless solid evidence or references were introduced at the beginning of the study, it would be really important to also assess the changes of immune cells in the pancreas.

Response: This suggestion is in line with a comment of Reviewer 1, who also suggested to perform a more detailed analysis of pancreatic immune infiltrate in CP mice (see comment 1/reviewer 1). In additional experiments, we isolated cells from the pancreas of CP mice as well as from untreated controls and analysed them by flow cytometry. Relatively few leukocytes could be measured in the pancreas of untreated animals, which correlates with our histological observations, and only CD206⁺ macrophages could be observed in significant amounts. The situation is different in CP animals: a marked increase of CCR2⁺ leukocytes suggests a prominent infiltration of leukocytes in CP. Mainly CD11b⁺, CD206⁺ or CD163⁺ M2 macrophages could be detected in the fibrotic tissue (**new Fig. 3 a-c**). In contrast to macrophages, CD11b⁺/Ly6g⁺ neutrophil granulocytes were detected in negligible numbers (**supplement fig. 2**). Beside CD206⁺ or CD163⁺ alternatively activated macrophages we also observed an increase of ILC2s (lin⁻/CD45⁺/CD127⁺/CD90⁺/Gata3⁺) in the pancreas of CP mice (**new Fig. 3f**). Infiltrating CD4⁺ as well as CD8⁺ T cells could also be detected in significant amounts (CD4⁺ cells in pancreas are shown in **new Fig. 2a-c**, CD8⁺ cells are analysed and shown in **new Fig. 2h**). A differentiated analysis of T-cell subpopulations in the pancreas (Treg, Th1, Th2 and Th17) is presented in **new Fig. 2d** of the revised manuscript. In addition to the different leukocyte populations, we detected pancreatic stellate cells by flow cytometry analysis via CD271 and GFAP labeling (**new Fig. 3h**). All these additional results were combined into **Figures 2 and 3** and were integrated into the revised manuscript. The new findings support our previous results and give evidence that the type 2 immune response in the pancreas involves not only CD206⁺ alternatively activated macrophages but also Th2-cells, ILC2s and Foxp3⁺ regulatory T-cells.

2. Since Tregs are known to be immunosuppressive and the balance of immune system may play crucial role in the physiological and pathophysiological condition. It would be of great importance to show the changes of T cell subtypes, such as CD4⁺ (Th1, Th2, and Th17), CD8⁺ etc in the spleen and possibly locally within the pancreas.

Response: The reviewer's concern addresses the question whether the deletion of the suppressive Tregs results in an undisturbed response of the T-effector cells and affects the balance of Th1/Th2 and Th17. In additional experiments we investigated T-effector cell subpopulations in spleen and in pancreatic tissue of C57Bl/6 wild type mice by a detailed flow cytometry analysis as well as by immunofluorescence labeling (**new Fig. 1 and Fig. 2**). In Treg depleted mice we isolated T-cells from spleen and analyzed their differentiation (Th1/Th2 and Th17) (**Fig.5d-f**). We observed an increase in Gata3⁺ Th2-cells whereas Tbet⁺ Th1-cells as well as Rorγt⁺ Th17-cells were not affected by the depletion of Tregs. CD4⁺ T-cells as well as CD8α⁺ T-cells were investigated in pancreatic tissue sections by immunofluorescent labeling (**Fig. 6a-c**). As suggested by the reviewer we also analyzed CD8⁺ cells in spleen and found an upregulation of CD8α⁺ T-cells following the depletion of Tregs. . Notably, it has already been reported that CD8α⁺ cells are associated with pro-fibrotic features in patients with systemic sclerosis [1] and lung fibrosis [2].

1. Fuschiotti P, Larregina AT, Ho J, Feghali-Bostwick C, Medsger TA Jr. Interleukin-13-producing CD8⁺ T cells mediate dermal fibrosis in patients with systemic sclerosis. IL-13-producing CD8⁺ T cells mediate dermal fibrosis in patients with systemic sclerosis Arthritis Rheum. 2013 Jan;65(1):236-46.
2. Brodeur TY, Robidoux TE, Weinstein JS, Craft J, Swain SL, Marshak-Rothstein A. IL-21 Promotes Pulmonary Fibrosis through the Induction of Profibrotic CD8⁺ T Cells. J Immunol. (2015) 195:5251–60.

3. Most of the results did not show important controls, specifically DEREK mice that were treated with PBS and DT without inducing chronic pancreatitis. Since it was reported Treg deficiency causes lethal and CD4⁺ T cell-driven autoimmune disease (PMID: 27994068).

Response: We thank the reviewer for this comment. It truly has been reported that Treg deficient mice can develop autoimmune disease patterns that are fatal [1]. Scurfy mice that have a natural knockout of the Treg essential transcription factor Foxp3 develop autoimmune disease shortly after birth [2]. For this reason, we used adult DEREK mice for our experiments, which have a normal Treg population. Whereas the neonatal depletion of Tregs by diphtheria toxin or depleting CD25 antibody results in a scurfy like disease, their depletion in adult mice does not [3]. As suggested by the reviewer we treated DEREK mice for 4 weeks with DT to examine effects which may interfere with pancreatitis and may cause the weight loss of the animals. Treatment with DT for 4 weeks did not result in weight loss or functional changes of the pancreas. Nevertheless, we observed in 2 out of 5 animals a beginning infiltration of T-cells in scattered areas of the pancreas. Serum lipase, which is a disease marker of acute pancreatitis, was not elevated in these mice. We did not observe any signs of acute pancreatitis or of autoimmune pancreatitis [4] 4 weeks after starting the Treg depletion. Furthermore we were also not able to detect any form of pancreas fibrosis or loss of exocrine tissue.

The new results are summarized in **Supplemental figure 6** and were included in the revised manuscript.

1. Sakaguchi S. Naturally arising CD4⁺ regulatory t cells for immunologic self-tolerance and negative control of immune responses *Annu Rev Immunol.* 2004;22:531-62.
2. Sharma R, Jarjour WN, Zheng L, Gaskin F, Fu SM, Ju ST. Large functional repertoire of regulatory T-cell suppressible autoimmune T cells in scurfy mice *J Autoimmun.* 2007 Aug;29(1):10-9.
3. Lahl K, Loddenkemper C, Drouin C, Freyer J, Arnason J, Eberl G, Hamann A, Wagner H, Huehn J, Sparwasser T. Selective depletion of Foxp3⁺ regulatory T cells induces a scurfy-like disease *J Exp Med.* 2007 Jan 22;204(1):57-63.
4. Schwaiger T, van den Brandt C, Fitzner B, Zaatreh S, Kraatz F, Dummer A, Nizze H, Evert M, Bröker BM, Brunner-Weinzierl MC, Wartmann T, Salem T, Lerch MM, Jaster R, Mayerle J. Autoimmune pancreatitis in MRL/Mp mice is a T cell-mediated disease responsive to cyclosporine A and rapamycin treatment *Gut.* 2014 Mar;63(3):494-505.

4. Also, it would be really helpful to provide the timeline for DT treatment related to the induction of chronic pancreatitis. Is there any organ-specific inflammation in those Tregs depleted mice before induction of chronic pancreatitis? What about depletion of Tregs about inducing chronic pancreatitis?

Response: We included an outline of the treatment scheme of DREG mice (**new supplemental fig. 1**). Furthermore, we investigated a possible organ specific inflammation in DREG mice 4 weeks after repetitive treatment with DT and in the absence of CP induction. We investigated pancreas, kidney, lung, liver and small intestine by H&E staining. As previously described it is known that CTLA4 deficient mice or Foxp3 deficient scurfy mice can develop lethal autoimmune disease. For both strains it is known that a general knockout resulted in severe autoimmune diseases, whereas the inducible knockout or depletion does not show such dramatic effects [1,2]. 4 weeks after starting Treg depletion we did not observe a severe inflammatory reaction in any of the investigated organs. In 2 out of 5 depleted animals we were able to see a beginning infiltration of T-cells in scattered areas of the pancreas (**new supplemental fig. 6**). Acute pancreatitis has been described in scurfy mice, but not in adult mice after depletion of regulatory T-cells. It has also been described that a constant depletion of Tregs by multiple injections of DT may trigger the rise of a resistant population of Tregs similar to what we observed in our model (**new supplemental fig. 5**). These results show that Treg depletion over 4 weeks in adult mice did not result in severe autoimmune diseases or the induction of chronic or acute pancreatitis.

1. Klocke K, Sakaguchi S, Holmdahl R, Wing K. Induction of autoimmune disease by deletion of CTLA-4 in mice in adulthood *Proc Natl Acad Sci U S A.* 2016 Apr 26;113(17):E2383-92.
2. Lahl K, Loddenkemper C, Drouin C, Freyer J, Arnason J, Eberl G, Hamann A, Wagner H, Huehn J, Sparwasser T. Selective depletion of Foxp3⁺ regulatory T cells induces a scurfy-like disease *J Exp Med.* 2007 Jan 22;204(1):57-63.

5. It is unclear the direct link between suppression of the splenic type 2 immune response and the progression of chronic pancreatitis. What happened after Treg depletion globally and locally during chronic pancreatitis?

Response: We agree with the reviewer that a splenic increase of Th2 cells would not necessarily induce pancreatic fibrosis. To address this issue, we performed flow cytometry analysis of leukocytes isolated from the chronic inflamed pancreas of mice. We observed that CD4⁺ T-cells infiltrate the organ, the majority of cells consisting of Foxp3⁺ Tregs and Gata3⁺ Th2-cells (**new Fig. 2a-d**). Whereas the splenic T-cell composition reflects the systemic inflammatory response of the chronic disease, the local T-cell infiltrate into the pancreatic tissue orchestrates specific intrapancreatic changes. To study the role of Th2-cells in the activation of alternative activated macrophage during CP, we blocked Th2 cells by using the CRTH2 antagonist OC000459. OC000459 inhibits the activation of Th2 cells [1] and their recruitment into the affected organ. Treated animals showed a significantly reduced infiltration of CD4⁺ T cells and a lower number of CD4⁺/Stat6⁺ Th2 cells in the pancreas. Also the number of CD206⁺ macrophages was also significantly reduced. Blockage of the Th2-M2 axis finally also reduced PSC activation, as shown by a diminished number of α SMA⁺ cells in the pancreas. Gene expression analysis by RT-qPCR of pancreatic tissue demonstrated significantly decreased transcript levels for *Mrc1*, *Col1a*, *Il4*, *Il10*, *Areg* and *Tgfb*, which confirmed the reduced type 2 immune response in OC000459 treated mice (**new Fig. 4a-j**).

By immunofluorescent labelling we could show that pancreatic CD4⁺ T-cells are the source of the M2 inducing cytokine IL13, and therefore directly trigger the M2 polarization. Labeling of T-cells in Treg depleted mice, showed an increase of CD4⁺ cells as well as of Stat6⁺ Th2-cells which also express IL13. Finally, the increased release of IL4 and IL13 results in an increased CD206⁺ M2 count within the pancreas and stimulates increased PSC activation and tissue fibrosis.

1. Pettipher R, Vinall SL, Xue L, Speight G, Townsend ER, Gazi L, Whelan CJ, Armer RE, Payton MA, Hunter MG. Pharmacologic profile of OC000459, a potent, selective, and orally active D prostanoid receptor 2 antagonist that inhibits mast cell-dependent activation of T helper 2 lymphocytes and eosinophils J Pharmacol Exp Ther. 2012 Feb;340(2):473-82.

6. It is unclear through which mechanism that Tregs modulates CD206⁺ macrophages or macrophage polarization? Why the authors specifically assessed CD206⁺ macrophages? What about other immune cells in the pancreas?

Response: *CD206* is a marker gene of M2-macrophages which are involved in the resolution of inflammatory responses and contribute to tissue repair and organ remodeling. The current literature shows that macrophages play essential roles in the course of both, acute and chronic pancreatitis [1,2]. They are involved in the tissue regeneration after an acute episode as well as in tissue fibrosis [2,3], which is a fundamental mechanism in chronic pancreatitis. Major drivers of M2 polarization are the cytokines IL4 and IL13 which are released by Th2-cells. T-cell activity, on the other hand, is counter regulated by Tregs which prevent excessive immune responses by immune suppression. In the following, M2 macrophages trigger the activation of pancreatic stellate cells which are important for the fibrotic remodelling of the pancreas [4, 5]. A role of M2 polarized macrophages as drivers of

fibrogenesis has been reported also in other fibro-inflammatory processes like liver fibrosis [6] or lung fibrosis [7]. CD206+ macrophages reflect the population of M2 macrophages in the pancreas [2]. As suggested by the reviewer we also investigated other immune cells in pancreatic tissue and integrated our results in the **new Fig. 2**. We observed an infiltration of CD4+ T-cells in the fibrotic pancreas, the major subclasses consisted of Foxp3+ Tregs and Gata3+ Th2-cells. Beside CD4+ T-cells we were able to detect CD8+ T-cells as well as ILC2s which could also affect tissue fibrogenesis [8, 9].

1. Wu J, Zhang L, Shi J, He R, Yang W, Habtezion A, Niu N, Lu P, Xue J. Macrophage phenotypic switch orchestrates the inflammation and repair/regeneration following acute pancreatitis injury EBioMedicine. 2020 Aug;58:102920.
2. Xue J, Sharma V, Hsieh MH, Chawla A, Murali R, Pandol SJ, Habtezion A. Alternatively activated macrophages promote pancreatic fibrosis in chronic pancreatitis Nat Commun. 2015 May 18;6:7158.
3. Criscimanna A, Coudriet GM, Gittes GK, Piganelli JD, Esni F. Activated macrophages create lineage-specific microenvironments for pancreatic acinar- and β -cell regeneration in mice Gastroenterology. 2014 Nov;147(5):1106-18.e11.
4. Apte MV, Haber PS, Darby SJ, Rodgers SC, McCaughan GW, Korsten MA, Pirola RC, Wilson JS. Pancreatic stellate cells are activated by proinflammatory cytokines: implications for pancreatic fibrogenesis Gut. 1999 Apr;44(4):534-41. doi: 10.1136/gut.44.4.534.
5. Mews P, Phillips P, Fahmy R, Korsten M, Pirola R, Wilson J, Apte M. Pancreatic stellate cells respond to inflammatory cytokines: potential role in chronic pancreatitis Gut. 2002 Apr;50(4):535-41. doi: 10.1136/gut.50.4.535.
6. Kisseleva T, Brenner D. Molecular and cellular mechanisms of liver fibrosis and its regression Nat Rev Gastroenterol Hepatol. 2021 Mar;18(3):151-166.
7. Borthwick LA, Barron L, Hart KM, Vannella KM, Thompson RW, Oland S, Cheever A, Sciruba J, Ramalingam TR, Fisher AJ, Wynn TA. Macrophages are critical to the maintenance of IL-13-dependent lung inflammation and fibrosis Mucosal Immunol. 2016 Jan;9(1):38-55.
8. Fuschiotti P, Larregina AT, Ho J, Feghali-Bostwick C, Medsger TA Jr. Interleukin-13-producing CD8+ T cells mediate dermal fibrosis in patients with systemic sclerosis. IL-13-producing CD8+ T cells mediate dermal fibrosis in patients with systemic sclerosis Arthritis Rheum. 2013 Jan;65(1):236-46.
9. Hams E, Armstrong ME, Barlow JL, Saunders SP, Schwartz C, Cooke G, Fahy RJ, Crotty TB, Hirani N, Flynn RJ, Voehringer D, McKenzie AN, Donnelly SC, Fallon PG. IL-25 and type 2 innate lymphoid cells induce pulmonary fibrosis Proc Natl Acad Sci U S A. 2014 Jan 7;111(1):367-72.

7. Since the focus of this study is the role of Tregs, it is difficult to understand why the author selected the growth factors from co-culture model of BMDM and CCK-stimulated acini. It would be really helpful to provide some evidence.

Response: It is well established that macrophages play an important role in organ remodeling [1,2]. Macrophages can activate pancreatic stellate cells via TGF β and thus drive organ fibrosis. However, macrophages not only produce TGF β , but also secrete other growth factors like inhibin/activin, insulin-like growth factor or NGF which could also contribute to fibrosis [3]. In the co-culture analysis we wanted to identify the specific growth factors that are released by macrophages during pancreatitis. Due to the technical difficulties of isolating functional macrophages from pancreatic tissue for transcriptome analysis, we decided to use instead an *in vitro* co-culture model where macrophages interact and react to the presence of necrotic acinar cells. The transcriptome analysis showed an upregulation of various growth factors (**Fig. 7a**) which were also found increased in Treg depleted mice (**Fig. 7b**). Our experiment suggests a regulation cascade where Tregs stimulate

macrophage differentiation via Th2-cells, and where macrophages regulate organ-remodeling by the secretion of specific growth factors which ultimately determine the fibrosis/regeneration balance.

1. Xue J, Sharma V, Hsieh MH, Chawla A, Murali R, Pandol SJ, Habtezion A. Alternatively activated macrophages promote pancreatic fibrosis in chronic pancreatitis *Nat Commun*. 2015 May 18;6:7158.
2. Wu J, Zhang L, Shi J, He R, Yang W, Habtezion A, Niu N, Lu P, Xue J. Macrophage phenotypic switch orchestrates the inflammation and repair/regeneration following acute pancreatitis injury *EBioMedicine*. 2020 Aug;58:102920.
3. Cruz AF, Rohban R, Esni F. Macrophages in the pancreas: Villains by circumstances, not necessarily by actions *Immun Inflamm Dis*. 2020 Dec;8(4):807-824.

8. Assessing the changes of ILC2 in the pancreas is interesting, however, it would be important to demonstrate and explain how Tregs potentially modulates ILC2.

Response: ILC2 are part of the type 2 immune response, and in our CP model we observed an upregulation of ILC2 in the same way as for Th2-cells. IL33 is a known inducer of ILC2 differentiation which acts via the ST2 receptor signaling pathway [1, 2]. Tregs are able to control ILC2 via the suppressive cytokines TGF- β and IL-10, but this also requires ICOS:ICOS-L interaction [3,4]. Our Treg-depletion experiments show an increased Type 2 immune response which goes along with an increased number of CD206+ macrophages in the pancreas (**new Fig. 3a-d**). These CD206+ macrophages are positive for the ILC2 inducing cytokine IL33 (**new Fig. 3e**). Apparently, the suppressive impact of Tregs on a Th2-cell differentiation also restrains a subsequent macrophage activation and ILC2 induction. All new results were included in the revised manuscript.

1. Riedel JH, Becker M, Kopp K, Düster M, Brix SR, Meyer-Schwesinger C, Kluth LA, Gnirck AC, Attar M, Krohn S, Fehse B, Stahl RAK, Panzer U, Turner JE. IL-33-Mediated Expansion of Type 2 Innate Lymphoid Cells Protects from Progressive Glomerulosclerosis *J Am Soc Nephrol*. 2017 Jul;28(7):2068-2080.
2. Takatori H, Makita S, Ito T, Matsuki A, Nakajima H. Regulatory Mechanisms of IL-33-ST2-Mediated Allergic Inflammation *Front Immunol*. 2018 Sep 4;9:2004.
3. Rigas D, Lewis G, Aron JL, Wang B, Banie H, Sankaranarayanan I, Galle-Treger L, Maazi H, Lo R, Freeman GJ, Sharpe AH, Soroosh P, Akbari O. Type 2 innate lymphoid cell suppression by regulatory T cells attenuates airway hyperreactivity and requires inducible T-cell costimulator-inducible T-cell costimulator ligand interaction *J Allergy Clin Immunol*. 2017 May;139(5):1468-1477.e2.
4. Maazi H, Patel N, Sankaranarayanan I, Suzuki Y, Rigas D, Soroosh P, Freeman GJ, Sharpe AH, Akbari O. ICOS:ICOS-ligand interaction is required for type 2 innate lymphoid cell function, homeostasis, and induction of airway hyperreactivity *Immunity*. 2015 Mar 17;42(3):538-51.

9. The resolution of the images needs to be improved in Figure 1a and g, Figure 3a and f, Figure 4c and e, Figure 5h and i, and Figure 6a.

Response: We increased the resolution of the Figures. In the original manuscript we included images in a lower resolution to minimise memory requirements. The revised version of the manuscript now contains tiff files of the individual figures which have a higher resolution.

10. Supplement Fig 1 is not for ILCs. Please check.

Response: The gating strategy is now summarised as supplemental figure 3 and integrated into the revised manuscript.

Reviewer #3 (Remarks to the Author):

Glaubitz et al explore immune control of chronic pancreatitis using a model induced by caerulein. The authors continue work published in a report last year. They document increased type 2 responses including Th2 cells and M2 macrophages. They then use the DERE model to demonstrate Treg-dependent control of Th2 responses, ILC2 expansion, fibrosis, and tissue destruction. Overall, the experiments are performed well and fairly convincing. However, there are few points to address. Moreover, the studies as shown document many correlations with Treg depletion, but lack clear mechanistic insight. This is discussed in specific comments.

1. There is a concern about some of the flow cytometry analysis. In Figure 1a, the immunofluorescence data suggests a dramatic increase in CD206+ macs. However, the flow cytometry shows a very modest difference and the CD206+ population is not even a clearly delineated population. Is there a possibility that the enzyme dissociation of the tissue is destroying the CD206 epitope? Could the IF be quantitated to provide this important piece of information?

Response: We thank the reviewer for this comment. The flow cytometry analysis showed only a minor increase of CD206⁺ M2 macrophages in spleen, but during chronic pancreatitis we observed a dramatic increase within the fibrotic pancreatic tissue. We now quantified CD206⁺ cells in sections of the whole pancreas and included the results in the **new Fig. 3d**. CD206⁺ macrophages are the dominating leukocyte population located in the pancreas of CP mice. Furthermore, we isolated leukocytes from pancreatic tissue and performed flow cytometry analysis (**new Fig. 3b and 3c**). The differences between flow cytometry analysis and histologic quantification could be explained by the loss of acinar cells during the process of tissue dissociation, which alters the percentage of total cells. Also, a partial desintegration of the epitope during isolation may have caused the lower detection by flow cytometry analysis. Notably, both analyses showed a marked increase of CD206+ M2 macrophages within the fibrotic pancreas. These results are in good agreement with current publications which state an essential role for pancreatic macrophages in organ remodelling [1-3].

1. Xue J, Sharma V, Hsieh MH, Chawla A, Murali R, Pandol SJ, Habtezion A. Alternatively activated macrophages promote pancreatic fibrosis in chronic pancreatitis *Nat Commun*. 2015 May 18;6:7158.
2. Wu J, Zhang L, Shi J, He R, Yang W, Habtezion A, Niu N, Lu P, Xue J. Macrophage phenotypic switch orchestrates the inflammation and repair/regeneration following acute pancreatitis injury *EBioMedicine*. 2020 Aug;58:102920.
3. Cruz AF, Rohban R, Esni F. Macrophages in the pancreas: Villains by circumstances, not necessarily by actions *Immun Inflamm Dis*. 2020 Dec;8(4):807-824.

2. Similar to the point above, the GATA3 staining in Fig. 1 shows a shift in the entire population. The Th2 population should be confirmed with another parameter; either IL-4 and/or IL13 expression or ST2 staining.

Response: To confirm the polarization of Gata3⁺ cells we isolated CD4⁺ T-cells from spleen and performed intracellular cytokine staining of IL4 and IL13. Increased numbers of IL4 and IL13 expressing T-cells were detected (**new Fig. 1h**). qRT-PCR analysis data also show enhanced gene expression of IL4 and IL13 (**new Fig. 1i**). These observations are further supported by increased serum levels of IL 4 in animals with chronic pancreatitis (**new Fig. 1g**). Furthermore we investigated T-cell populations in chronic pancreatitis tissue and observed increased ratios of Gata3⁺ Th2-cells (**new Fig. 2d**). By immunofluorescent labelling we identified IL13 producing CD4⁺ T-cells in pancreatic tissue (**new Fig. 2f**). CD294 (CRTH2), is a chemoattractant receptor-homologous molecule expressed on Th2 cells [1]. We used CD294 as a second marker of Th2 cells in flow cytometry analysis and also observed an increase of the Th2-cell population (**new Fig. 2g**). All new results were included in the revised manuscript.

1. Cosmi L, Annunziato F, Galli MIG, Maggi RME, Nagata K, Romagnani S. CRTH2 is the most reliable marker for the detection of circulating human type 2 Th and type 2 T cytotoxic cells in health and disease *Eur J Immunol.* 2000 Oct;30(10):2972-9.

3. There are many correlations following Treg depletion (increased Th2, increased M2, increased ILC2, altered cytokine profiles, increased fibrosis and tissue destruction, weight loss). However, there are no experiments functionally linking these observations to the pancreatitis that directly support the conclusion(s) in the title and abstract. The simplest would be to treat mice with anti-IL-4 to show that IL-4 is critical for the effects of Treg depletion. This would also determine if the changes observed in Fig. 4 and 6, which are less associated with IL-4 biology, are due to IL-4 or another pathway. On the point, the authors should be more careful about making causal conclusions with only correlative data.

Response: We agree and thank the reviewer for this suggestion. The IL4/IL13 axis is essential for the activation of CD206⁺ macrophages and the subsequent fibrotic tissue reorganization during CP. It is well established that IL4 triggered CD206⁺ macrophages activate pancreatic stellate cells (PSC). Xue et al reported that IL4/IL13^{-/-} mice, as well as IL4R α ^{-/-} mice, show reduced PSC activation and have reduced collagen production. In *Lyzm^{Cre} fl/fl IL4R α ^{-/-}* mice, which carry a macrophage-specific IL4R α knock out, they could demonstrate the essential role of macrophages in the IL4 signaling pathway. A blockage of the IL4/IL13 pathway by application of IL4/IL13 binding protein was shown to limit fibrosis in these animals [1]. To demonstrate that Tregs regulate fibrosis via the Th2-M2 axis we performed a similar approach by a therapeutic blockade of the Th2 response. We used a CRTH2 (chemoattractant receptor-homologous molecule expressed on Th2 cells) -specific inhibitor (OC000459) to block the transmigration of Th2-cells into the pancreas [2, 3]. The blockage of CRTH2 by OC000459 resulted in a decreased infiltration of T-cells in general, but especially of Th2-cells (CD4⁺/STAT6⁺). As a consequence we observed decreased numbers of CD206⁺ macrophages in the pancreas which was associated with the development of less tissue fibrosis. qRT-PCR analysis gave

evidence that IL4 transcripts are reduced in the OC000459 treated mice. The results from the treatment with OC000459 were included in the revised manuscript and are summarized in **Figure 4**.

1. Xue J, Sharma V, Hsieh MH, Chawla A, Murali R, Pandol SJ, Habtezion A. Alternatively activated macrophages promote pancreatic fibrosis in chronic pancreatitis Nat Commun. 2015 May 18;6:7158.
2. Cosmi L, Annunziato F, Galli MIG, Maggi RME, Nagata K, Romagnani S. CRTH2 is the most reliable marker for the detection of circulating human type 2 Th and type 2 T cytotoxic cells in health and disease Eur J Immunol. 2000 Oct;30(10):2972-9.
3. Pettipher R, Vinall SL, Xue L, Speight G, Townsend ER, Gazi L, Whelan CJ, Armer RE, Payton MA, Hunter MG. Pharmacologic profile of OC000459, a potent, selective, and orally active D prostanoid receptor 2 antagonist that inhibits mast cell-dependent activation of T helper 2 lymphocytes and eosinophils J Pharmacol Exp Ther. 2012 Feb;340(2):473-82.

4. The CD90 IF in Fig. 5h and I is problematic because CD90 is not restricted to ILCs. It is also expressed on T cells. The authors need to distinguish these populations.

Response: It is true that CD90 is expressed on many immune cells and amphiregulin is also produced by Th2 cells. We included the IF data in Fig.5h and i (**new Fig. 8h and i**) only to envision the flow cytometry data we presented in Fig. 5 e and f (**new Fig. 8e and f**). Flow cytometry analysis allowed a more detailed differentiation of the cell types. In flow cytometry we identified ILCs as lin⁻/CD45⁺/CD127⁺/CD90⁺ and afterwards detected ILC2 by gating Gata3⁺ ILCs. As ILCs do not express CD4, we can exclude that these cells are Gata3⁺/CD4⁺ Th2 cells.

Figure for reviewer only: CD8α⁺ and CD4⁺ cells were excluded and only CD8α⁻/CD4⁻ were used for further gating of ILCs, finally we could exclude CD90⁺/CD4⁺ cells from the population of ILCs.

5. Since the authors have access to patient samples (Fig. 5i), have they stained for CD4/Foxp3 to show that Tregs are present in CP patient biopsies? This would seem to be central to the report's conclusions that Tregs limit the CP inflammation and disease.

Response: As we had limited access to patient material from resected chronic pancreatitis patients (ChroPac trial (ISRCTN38973832) [1]), we followed the reviewers suggestion and stained paraffin embedded tissue samples for CD4 and Foxp3. We identified a high number of Foxp3+/CD4+ cells in the pancreatic tissue of CP patients (**new Fig 10e**). Additionally, we collected blood samples from CP patients and healthy blood donors for flow cytometry analysis of circulating Tregs and effector T-cells. We observed significantly increased numbers of Foxp3+/CD25+ Tregs as well as Gata3+ Th2-cells in patients with CP (**new. Fig. 10a-d**). Increased systemic Treg and Th2-cells as well as increased local Treg and Th2 infiltration in patients suffering from chronic disease equally match our observations in the CP mouse model. The new data from patient samples were included in the revised manuscript (**new Fig. 10**).

1. Diener MK, Hüttner FJ, Kieser M, Knebel P, Dörr-Harim C, Distler M, Grützmann R, Wittel UA, Schirren R, Hau HM, Kleespies A, Heidecke CD, Tomazic A, Halloran CM, Wilhelm TJ, Bahra M, Beckurts T, Börner T, Glanemann M, Steger U, Treitschke F, Staib L, Thelen K, Bruckner T, Mihaljevic AL, Werner J, Ulrich A, Hackert T, Büchler MW; ChroPac Trial Group. Partial pancreatoduodenectomy versus duodenum-preserving pancreatic head resection in chronic pancreatitis: the multicentre, randomised, controlled, double-blind ChroPac trial. *Lancet*. 2017 Sep 9;390(10099):1027-1037.

REVIEWER COMMENTS

Reviewer #1 (Remarks to the Author):

Author responses are adequate and no additional changes are needed.

Reviewer #2 (Remarks to the Author):

The authors have fully addressed my comments and done great job on revising the manuscript, I do not have further questions.

Reviewer #3 (Remarks to the Author):

The authors have done a reasonably good job of addressing some of the previous comments. There has been substantial new data added to the revised report. However, there are a couple of issues raised with some of the responses.

1. The authors have tried to address the question about the source of IL-4 and IL-13. They present evidence that Th2-like cells are positive for those cytokines but then seem to assume that these cells are the only source. They make a separate point that ILC2 (Gata3+) are expanded in CP. The authors need to directly examine ILC2s in the system.

2. The experiments in Figure 4 are extremely problematic. Although this experiment was performed to address comments from multiple reviewers it raises more questions than it addresses. First, the OC000459 compound, from my read of the literature, has never been used in mouse models. Part of the reason for that is that CRTH2 does not show the restricted expression in murine systems that it does in humans. In fact, deficiency in the prostaglandin D2 receptor in mice has a modest phenotype. So while the data showing the inhibitor has an effect are reasonable, the mechanism is unknown and certainly not what the authors are concluding.

3. A minor point but in the last panel of Fig 10, the staining with STAT6 is meaningless. STAT6 is broadly expressed in T cells and other cells. It is NOT specific for Th2 cells. This panel could be deleted.

REVIEWER COMMENTS

Reviewer #1 (Remarks to the Author):

Author responses are adequate and no additional changes are needed.

Response: Thank you.

Reviewer #2 (Remarks to the Author):

The authors have fully addressed my comments and done great job on revising the manuscript, I do not have further questions.

Response: Thank you.

Reviewer #3 (Remarks to the Author):

The authors have done a reasonably good job of addressing some of the previous comments. There has been substantial new data added to the revised report. However, there are a couple of issues raised with some of the responses.

1. The authors have tried to address the question about the source of IL-4 and IL-13. They present evidence that Th2-like cells are positive for those cytokines but then seem to assume that these cells are the only source. They make a separate point that ILC2 (Gata3⁺) are expanded in CP. The authors need to directly examine ILC2s in the system.

Response: Several reports from the literature have shown that ILC2s are the source of the cytokines IL4 and IL13 [1-3]. In additional flow cytometry experiments, we could also directly show that Gata3⁺ (lin⁻ CD45⁺/CD127⁺/CD90⁺) ILC2s isolated from the spleen express cytokines IL-4 and IL-13. Furthermore, we performed immunofluorescent labelling of Gata3, CD4 and the cytokines IL-4 and IL-13 in chronic pancreatitis tissue samples to identify Gata3⁺/CD4⁻ cells which express cytokines IL-4 and IL-13. These new data which show that ILC2s in spleen as well as in CP tissue are a source of the cytokines IL-4 and IL-13 were included in the manuscript (**supplement. fig. 4a-b**).

1. Pelly VS, Kannan Y, Coomes SM, Entwistle LJ, Rückerl D, Seddon B, MacDonald AS, McKenzie A, Wilson MS. IL-4-producing ILC2s are required for the differentiation of T_H2 cells following *Heligmosomoides polygyrus* infection Mucosal Immunol. 2016 Nov;9(6):1407-1417.
2. Halim TY, Steer CA, Mathä L, Gold MJ, Martinez-Gonzalez I, McNagny KM, McKenzie AN, Takei F. Group 2 Innate Lymphoid Cells Are Critical for the Initiation of Adaptive T Helper 2 Cell-Mediated Allergic Lung Inflammation Immunity. 2014 Mar 20;40(3):425-35.
3. Hams E, Armstrong ME, Barlow JL, Saunders SP, Schwartz C, Cooke G, Fahy RJ, Crotty TB, Hirani N, Flynn RJ, Voehringer D, McKenzie AN, Donnelly SC, Fallon PG. IL-25 and type 2 innate lymphoid cells induce pulmonary fibrosis Proc Natl Acad Sci U S A. 2014 Jan 7;111(1):367-72.

2. The experiments in Figure 4 are extremely problematic. Although this experiment was performed to address comments from multiple reviewers it raises more questions than it addresses. First, the OC000459 compound, from my read of the literature, has never been used in mouse models. Part of the reason for that is that CRTH2 does not show the restricted expression in murine systems that it does in humans. In fact, deficiency in the prostaglandin D2 receptor in mice has a modest phenotype. So while the data showing the inhibitor has an effect are reasonable, the mechanism is unknown and certainly not what the authors are concluding.

Response: In order to address the reviewer comments to investigate the Th2 M2 axis we decided to treat the mice with the CRTH2 antagonist OC000459. CRTH2 is expressed on human Th2 cells [1] and is able to regulate Th2-cell migration [2]. Also in the murine system, CRTH2 has been shown to be expressed on CD4⁺ cells and to control a Th2 immune response [3]. The *in vivo* inhibition of CRTH2 by the inhibitor CAY10595 showed a reduced T-cell infiltration and stellate cell activation in a mouse model of pulmonary hypertension [3]. Furthermore, Wojno et al used the CRTH2 antagonist OC000459 in a mouse model of chronic inflammatory lung disease to block ILC2 accumulation [4]. According to these publications we used OC000459 to block Th2/ILC2 response during chronic pancreatitis to prove the hypothesis that IL4/IL13 expression in Th2-cells and ILC2s contributes to M2 polarisation and fibrogenesis. We decided to use the OC000459 antagonist because it is already being used in clinical trials to treat allergic/fibrotic lung diseases and therefore represents a therapeutic option for the treatment of patients [5]. Nevertheless, we agree with the reviewer that CRTH2 is not exclusively expressed on Th2 cells. To give our observation further evidence we performed additional experiments to investigate the effect of the IL-4 signalling during our model of chronic pancreatitis. IL-4 is mainly involved in the differentiation of Gata3⁺ Th2-cells [6]. To further analyse the IL-4 pathway we used neutralizing anti-IL-4 antibodies in our mouse model of CP, the treatment scheme is shown in **supplementary figure 1b**. We observed a decreased type 2 immune response after anti-IL-4 treatment, with significantly reduced Gata3⁺ Th2-cells as well as Gata3⁺ (lin⁻ CD45⁺/CD127⁺/CD90⁻) ILC2s (**supplementary figure 5b and 5c**). Also, in pancreatic tissue we found a diminished infiltration of CD4⁺ T-cells, and especially STAT6⁺/CD4⁺ T-cells were completely absent in the anti-IL-4 treated mice (**supplementary figure 5d and 5e**). The lack of a Th2-response finally resulted in a significantly reduced fibrosis and stellate cell activation, as shown by Masson Goldner and azan blue staining, and also by immunofluorescent labelling of α -SMA and collagen 1 (**supplementary figure 5f-h**). IL-4 clearly contributes to pancreatic fibrogenesis in the context of CP. Our results are in line with previously published data by the group of Habtezion et al who could show that a blockage of the IL-4/IL-13 receptor signalling pathway by using an IL-4/IL-13 blocking peptide resulted in a diminished pancreatic stellate cell activation and reduced fibrosis [7]. In summary both treatments, with OC000459 as well as with the anti-IL-4 antibody, resulted in diminished pancreatic T-cell infiltration and reduced pancreas fibrogenesis which suggests a critical role for the Th2 and ILC2 response for organ remodelling during CP. Furthermore, we could show that Th2-cells as well as ILC2s are

a source of cytokines IL-4 and IL-13, which are known to drive alternative macrophage activation and tissue fibrosis.

1. Cosmi L, Annunziato F, Galli MIG, Maggi RME, Nagata K, Romagnani S. CRTH2 is the most reliable marker for the detection of circulating human type 2 Th and type 2 T cytotoxic cells in health and disease *Eur J Immunol*. 2000 Oct;30(10):2972-9.
2. Hirai H, Tanaka K, Yoshie O, Ogawa K, Kenmotsu K, Takamori Y, Ichimasa M, Sugamura K, Nakamura M, Takano S, Nagata K. Prostaglandin D₂ selectively induces chemotaxis in T helper type 2 cells, eosinophils, and basophils via seven-transmembrane receptor CRTH2 *J Exp Med*. 2001 Jan 15;193(2):255-61.
3. Chen G, Zuo S, Tang J, Zuo C, Jia D, Liu Q, Liu G, Zhu Q, Wang Y, Zhang J, Shen Y, Chen D, Yuan P, Qin Z, Ruan C, Ye J, Wang XJ, Zhou Y, Gao P, Zhang P, Liu J, Jing ZC, Lu A, Yu Y. Inhibition of CRTH2-mediated Th2 activation attenuates pulmonary hypertension in mice *J Exp Med*. 2018 Aug 6;215(8):2175-2195.
4. Wojno ED, Monticelli LA, Tran SV, Alenghat T, Osborne LC, Thome JJ, Willis C, Budelsky A, Farber DL, Artis D. The prostaglandin D₂ receptor CRTH2 regulates accumulation of group 2 innate lymphoid cells in the inflamed lung *Mucosal Immunol*. 2015 Nov;8(6):1313-23.
5. Barnes N, Pavord I, Chuchalin A, Bell J, Hunter M, Lewis T, Parker D, Payton M, Collins LP, Pettipher R, Steiner J, Perkins CM. A randomized, double-blind, placebo-controlled study of the CRTH2 antagonist OC000459 in moderate persistent asthma *Clin Exp Allergy*. 2012 Jan;42(1):38-48.
6. Kaplan MH, Schindler U, Smiley ST, Grusby MJ. Stat6 is required for mediating responses to IL-4 and for development of Th2 cells *Immunity*. 1996 Mar;4(3):313-9.
7. Xue J, Sharma V, Hsieh MH, Chawla A, Murali R, Pandol SJ, Habtezion A. Alternatively activated macrophages promote pancreatic fibrosis in chronic pancreatitis *Nat Commun*. 2015 May 18;6:7158.

3. A minor point but in the last panel of Fig 10, the staining with STAT6 is meaningless. STAT6 is broadly expressed in T cells and other cells. It is NOT specific for Th2 cells. This panel could be deleted.

Response: We thank the reviewer for this point. We replaced the panel by a schematic illustration to visualise how Tregs interact in the course of chronic pancreatitis.

REVIEWERS' COMMENTS

Reviewer #3 (Remarks to the Author):

I thank the authors for the additional data. They have greatly strengthened their report.

REVIEWER COMMENTS

Reviewer #3 (Remarks to the Author):

I thank the authors for the additional data. They have greatly strengthened their report.

Response: Thank you.